# Supervised Distributed Multi-Instance and Unsupervised Single-Instance Autoencoder Machine Learning for Damage Diagnostics with High-Dimensional Data—A Hybrid Approach and Comparison Study

Stefan Bosse [1,*] , Dennis Weiss [1] and Daniel Schmidt [2]

1 Department Mathematics & Computer Science, University of Bremen, 28359 Bremen, Germany; Dennis.Weiss3@web.de
2 German Aerospace Center (DLR e. V.), Institute of Composite Structures and Adaptive Systems, 38108 Braunschweig, Germany; Daniel.Schmidt@dlr.de
* Correspondence: sbosse@uni-bremen.de

**Abstract:** Structural health monitoring (SHM) is a promising technique for in-service inspection of technical structures in a broad field of applications in order to reduce maintenance efforts as well as the overall structural weight. SHM is basically an inverse problem deriving physical properties such as damages or material inhomogeneity (target features) from sensor data. Often models defining the relationship between predictable features and sensors are required but not available. The main objective of this work is the investigation of model-free distributed machine learning (DML) for damage diagnostics under resource and failure constraints by using multi-instance ensemble and model fusion strategies and featuring improved scaling and stability compared with centralised single-instance approaches. The diagnostic system delivers two features: A binary damage classification (damaged or non-damaged) and an estimation of the spatial damage position in case of a damaged structure. The proposed damage diagnostics architecture should be able to be used in low-resource sensor networks with soft real-time capabilities. Two different machine learning methodologies and architectures are evaluated and compared posing low- and high-resolution sensor processing for low- and high-resolution damage diagnostics, i.e., a dedicated supervised trained low-resource and an unsupervised trained high-resource deep learning approach, respectively. In both architectures state-based recurrent artificial neural networks are used that process spatially and time-resolved sensor data from experimental ultrasonic guided wave measurements of a hybrid material (carbon fibre laminate) plate with pseudo defects. Finally, both architectures can be fused to a hybrid architecture with improved damage detection accuracy and reliability. An extensive evaluation of the damage prediction by both systems shows high reliability and accuracy of damage detection and localisation, even by the distributed multi-instance architecture with a resolution in the order of the sensor distance.

**Keywords:** structural health monitoring; distributed sensor networks; distributed machine learning; model fusion; autoencoder learning

## 1. Introduction and Related Work

Structural health monitoring (SHM) based on Lamb waves, a type of ultrasonic guided waves, is a promising technique for in-service inspection of aircraft structures. The implementation of SHM systems into aircraft applications reduces maintenance efforts as well as overall structural weight. Lamb waves are excited and received using a network of actuators and sensors, which are permanently attached on the structure. Lamb waves are very sensitive and exhibit different wave interaction mechanisms, with structural damages, such as attenuation, reflection, scattering or mode conversion. By analysing the sensor signals, different kinds of structural damages can be detected and located [1,2].

Automatic and reliable damage diagnostics using SHM systems is still a challenge, especially in the case of carbon fibre laminate due to their anisotropic material characteristics. Depending on the underlying measuring technique used to retrieve suitable sensor signals that show a sufficient correlation with damage or fatigue features, the recognition of the damage features requires complex analysis with experts knowledge and intervention [3]. Moreover, damage diagnostics can be an inherently distributed problem [4] using spatially distributed sensors [5] still processed by a central instance leading to scaling and efficiency issues. Scaling is limited with such centralised architectures. However, distributed data processing in sensor networks, especially addressing material-applied or material-integrated sensor networks, imposes strict resources constraints of the signal processors both regarding memory and computational power of each unit.

Damage and structural health diagnostics is an inverse problem. A model $M$ represents a measurement that maps a spatial and time-dependent environmental context $p_e(x,t)$ with a feature set $f$ (e.g., damage class and location) of a device under test (DUT) on sensor signal data $s$. The damage diagnostics system requires the inverse model $M^{-1}$ that maps the sensor data on the requested features to be monitored (related to another measuring parameter set $p_m$):

$$M(\vec{x},t,\vec{p}_e,\vec{f}) : (\vec{x},\vec{p},\vec{f}) \rightarrow \vec{s}$$
$$M^{-1}(\vec{s},\vec{p}_m) : \vec{s} \rightarrow \vec{f} \tag{1}$$

Beside numerical methods (e.g., inverse numeric [6]), machine learning (ML) can be utilised to derive the inverse model $M^{-1}$ from training example data mapping $s$ on $f$. Due to the highly non-linear model function artificial neural networks (ANN) are often used to implement a hypothesis of the required damage predictor function [7].

The task of structural health monitoring systems is to detect and locate different kind of damages from sensor data which are produced by permanent applied sensors on the structure. Related work can be classified in model-based [8] and model-free methods. Damage diagnostics with homogeneous and isotropic materials, e.g., aluminium or steel, can be handled with established methods. However, dealing with materials posing complex physical relationship between damages and sensor signals, e.g., by anisotropic and non-linear interaction behaviour, such as in composite laminates, deriving suitable models that map sensor data on state information is a still challenge or still not possible.

The main objective of this work is the investigation of model-free distributed machine learning (DML) under resource and failure constraints (including sensor noise, drift, and fatigue) by using spatial model decomposition and global model fusion strategies. Distributed learning and feature inference gains attraction in recent years to overcome scaling and reliability issues, originally applied in wireless structural health monitoring networks [5]. Additionally, state-based ML should process time-resolved sensor data, already successfully applied in the field of SHM with guided waves [9]. The damage diagnostic system is operating in space and time dimension.

Commonly, low-resource and low-resolution approaches used in damage monitoring that can be deployed on embedded computers typical for sensor networks pose only limited and operational constrained damage diagnosis capabilities (with respect to classification and localisation of damages), whereas high-resolution approaches require high computational time and storage requirements, often utilising deep learning and computer vision (CV) [10]. Deep learning of artificial neural networks (ANN) are often used for SHM [11]. Base-line approaches either try to derive damage feature by analysing differences between a non-damaged base-line experiment and a device under test or by using more advanced approaches together with ML. Auto-encoders (AE) are candidates to detect anomalies in sensor signals related to damage features that can be derived with deep Learning, basically by modelling multiple levels of visual abstraction (from low-level features to higher-order representations, i.e., features of features) from the sensor data [11]. In a first step an AE approach encodes a signal. A second step reconstructs (decodes) the signal again. If the AE is trained with ground truth data only it will not be able to reconstruct a signal containing

differences due to anomalies (the feature to be detected), e.g., a damage that modifies a sensor signal. Comparing the reconstructed signals with the original signal enables damage detection without a supervised training with labelled data.

The proposed ML architecture should be suitable for processing on low-resource embedded computers such as material-integrated sensor nodes of a sensor network [12]. Ideally, a node of the sensor network processes only local sensor data and performs local damage diagnostics. The damage predictor functions have to be highly discriminative with respect to noise and varying operational and measuring conditions [13]. There are two levels of prediction, i.e., feature extraction from the sensor signals: 1. The classification of damage and non-damage cases; and 2. The prediction of the spatial properties. In principle, there is a third level that classifies the damage class and its cause. This level is not addressed in this work.

Typically, the predictor functions are derived by supervised learning and sensor data derived from experiments with a defined and fixed parameter set of specimen geometry, actuator configuration and measurement setup, i.e., excitation frequencies, frequency filters, sensor scan grid as well as damage type and position [13]. Operational variance (e.g., temperature and humidity) have to be considered, too (contained in the parameter sets $p_e$ and $p_m$). One major challenge is training of the predictor functions with limited variance of training data, concerning the variance of experiments of a single set-up to cover typical measuring and specimen variations (i.e., repetitions of experiments under same conditions with the same parameter set for the device under test, damage, and the measuring set-up) and the variance of experiments and the respective features (i.e., different damage cases, classes, positions, sensors, environmental conditions). This limited training data results commonly in a lack of required generalisation of the prediction model that cannot be transferred to a broader range of parameter sets and unknown specimen configurations.

Two different approaches are compared in this work, which are finally fused to a hybrid system: A multi-instance low-resolution and a single-instance high-resolution architecture differing in resource requirements and the training class (supervised versa unsupervised learning, respectively). Common to both approaches is the deployment of state-based recurrent ANN (RNN) processing time-resolved sensor signal data from a spatially bounded context (i.e., local sensor data processing). The low-resolution approach should be capable to be used in structural applied (material-integrated) sensor networks (i.e., SHM at run-time), whereas the high-resolution approach can be primarily used for laboratory diagnosis or at service-time. On one hand the high-resolution approach delivers an assessment base for the low-resolution approach, on the other side the low-resolution approach can be used as a fast approximating region-of-interest feature marker for the high-resolution system.

Given recent advances in sensor technologies and micro-system integration this article proposes that a robust, yet simpler, real-time capable and low-resource, distributed machine learning approach is now available for accurately estimating damages in hybrid materials compared to conventional sensor analysis and deep learning approaches.

SHM based on Lamb waves and ultrasonic measuring techniques enables the detection of different kinds of structural damages and their localisation [1,2]. However, the presence of at least two Lamb wave modes (symmetric modes, $S_0, S_1, S_2, \ldots$, and anti-symmetric modes, $A_0, A_1, A_2, \ldots$) at any given frequency, their dispersive characteristic and their interference at structural discontinuities produce complex wave propagation fields. Due to the complex wave fields, conventional algorithms are reaching their limits for robust damage detection and localisation with the application.

In order to develop new damage detection algorithms based on machine learning the experimental air-coupled ultrasonic technique is used. With this technique the Lamb wave propagation field can be measured at any position of the structure. The measured wave propagation at a given position is used as sensor data for damage detection. The machine learning approaches require high amount of experimental data sets with different damage types and locations. Therefore, different removable pseudo defects are developed

which can be applied to different locations of the structure and generates comparable wave interactions such as real structural damages.

In the next sections the basic requirements for signal data processing and ML are presented, including a description of the origin of the sensor data and the physics of wave propagation relevant to understand damage detection. Furthermore, the present paper contains two main sections, one for each ML architecture and training approach. Finally, both approaches are compared and fused to a hybrid architecture (although, more as an outlook).

## 2. Machine Learning and Sensor Networks

### 2.1. Feature Selection and Extraction

Feature selection and extraction is the process to derive meaningful information related to a target variable $y$ from sensor data related to the observation variable $x$. Therefore, any feature selection can be represented by a generalised function $\Omega(s): s \to f$. There are input data features and target variable features to be distinguished.

The process of input feature selection is typically related to sensor data pre-processing that transforms and reduces the raw sensor data $s$ to relevant information $s_f = x$ contained in the signal $s$ with respect to target variable $y$ (defining the input vector $x$), e.g., using time-frequency transformation to get selected frequencies from the signal, the variance of the signal or other signal features.

In this work there are spatial and temporal relevant features that have to be selected to perform the final feature extraction that delivers damage feature vector $F = \langle \mathcal{D}, \mathcal{P} \rangle$ (categorical damage classification $\mathcal{D}$ and estimation of the spatial position of the damage $\mathcal{P}$), related to the target variable $y$. The signal feature selection is performed in this work primarily by a wavelet analysis of the time-resolved sensor signals, discussed in Section 5.2.

The target output feature extraction (damage classification and estimation of the spatial position of the damage) is then performed by the model function $M^{-1}$ introduced at the beginning and derived by machine learning (using the pre-processed input data features).

### 2.2. Taxonomy of Architectures

It is assumed that there is a sensor network $SN$ represented by a graph $G = \langle \mathcal{S}, \mathcal{C} \rangle$ that consists of numbered $(i,j)$ and identifiable sensor nodes $S(p)_{i,j} \in \mathcal{S}$ each at a different spatial position $p = (x,y)$ providing at least one time-resolved sensor signal $s(t)$. The sensor nodes can communicate with each other via a network structure (vertices of the $SN$ graph) with connections $com_{ij,kl} \in \mathcal{C}$.

There are basically two main strategies of sensor data aggregation and sensor network architectures for machine learning that can be deployed to derive a feature vector $F$ of the device under test (DUT) from the sensor data matrix $D$, i.e., the target variable $y$ related to the global state $ST$ of the DUT:

**Global Learning**

Inference by and training of a single predictive model instance $M$ using a spatially collected data record series $D(t)$ sampled at a certain time $t$ (or averaged over a time interval) that is processed by a central processing instance (one processing node). The single instance directly delivers the global feature vector $F$ related to the global state $ST$.

$$D(t) = \begin{bmatrix} d_{1,1} & \cdots & d_{n,1} \\ \vdots & \ddots & \vdots \\ d_{1,m} & \cdots & d_{n,m} \end{bmatrix}, M(D) : D \to y \qquad (2)$$

**Local Learning with Global Fusion**

This is the inference by and training of $n \times m$ multiple predictive model instances $\mu_{i,j}$ (distributed model ensemble) related to sensor nodes $S_{i,j}$ with local sensor data processing. Time-resolved local sensor data series $d_{i,j}(t)$ at a specific spatial position $p(S)$ of node $S_{i,j}$ is processed by each instance of the ensemble locally and independently. Each instance

estimates a local state *st* and delivers a local feature vector $f(p)_{i,j}$. All local states are finally fused to a global state *ST* delivering a global feature vector *F* by a fusion function $\Phi(\mu_{1,1}, \mu_{1,2}, \dots, \mu_{n,m})$ with index *n* and *m* as the number of nodes in each dimension.

$$d_{i,j}(t) = \left[s_{i,j}(t_0), \dots, s_{i,j}(t)\right], \mu_{i,j}(d_{i,j}) : d_{i,j} \to y_{i,j}, M = \bigcap_{i,j}\mu_{i,j} \qquad (3)$$

Both architectures are compared in Figure 1. Global fusion of a predictor function estimating the global state from an ensemble of local prediction or classification models related to a local state can be performed by probalistic methods, negotiation, majority election, and in the simplest case by spatial averaging. The fusion strategies used in this work are discussed in the Sections 4 and 5.

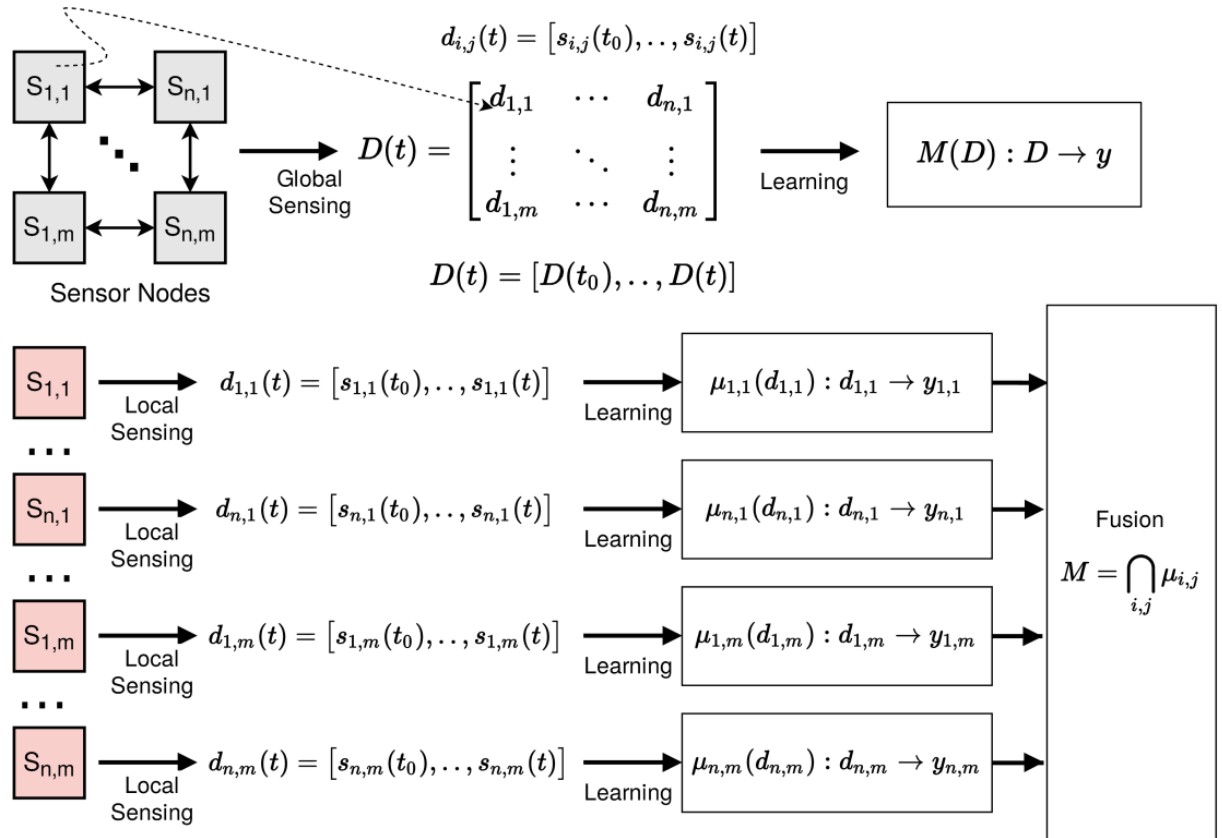

**Figure 1.** Spatial vs. temporal dimensions of the sensor data and centralised vs. distributed sensor processing and ML. (**Top**) Centralised data sampling and one global model *M*. (**Bottom**) Decentralised data sampling with local models $\mu$.

These two learning architectures can be further classified in training and application sub-architectures, leading to a classification scheme based on multi-instance (M) and single-instance (S) training (T) and prediction/inference (P) classes (similar to Flynn's computer architecture taxonomy based on stream classification):

**STSP**

A single learning instance with global input data processing and global output state prediction is used for training and prediction.

**MTMP**

Multiple learning instances with local input data processing and local output state prediction are used for training and prediction. This is the first ML architecture class considered in this work. The global state prediction is performed by fusion of the output of the individual learning instances.

**STMP**

A single learning instance with global input data processing but local output state prediction is used for training and replicated multiple instances with local input data are used for application. This is the second architecture considered in this work.

**MTSP**

Multiple learning instances with local input data processing and local output state prediction finally fused to one global model instance.

Fusion can be classified in input data, model, and output data fusion. Input data fusion is used typically on a global learning (STSP) level, model fusion is the transformation and reduction of multiple trained models to one generalised global model (MTSP), and output data fusion is the fusion of the multiple local state predictions to one global state prediction (MTMP/STMP). Model fusion can be used to increase the prediction accuracy and/or to combine local prediction models to one global prediction model.

There is no sensor interaction in terms of communication; the distributed sensor signals are correlated by the wave propagation. A single sensor node processes only its local sensor and passes the pre-processed data to its local learning instance only predicting the local state (e.g., a damage nearby), discussed in the next sections.

*2.3. Generalization*

With respect to supervised learning, training data $T = \langle D, Y \rangle$ is used for learning (sensor data with target variable association). Testing a trained model is performed by statistical error analysis of:

1. The training data;
2. Test data not used for training; and
3. Training and test data combined sets.

Therefore, experimental and simulation data must be split in training and test sets. The test has to evaluate:

- Accuracy of prediction;
- False-positive and false-negative error rates;
- "Offset" and distortion.

A major problem with machine learning is the tendency to a weak generalization of the model, i.e., only a specialised model was derived:

- The learned model can accurately represent the training data, but not the test data (special model)
- The learned model depends on geometric or temporal variables, such as the measuring location or signal phase/time offset
- The prediction region, i.e., local vs. global models.

The cross-validation test can be used during the training phase to adapt training parameters, to select a sub-set of model instances trained separately and with Monte Carlo simulation in case of single-instance learning, and in the case of multi-instance learning to select bad instances for progressive post-training.

*2.4. Sensor Processing*

The sensor data processing and data flow consists of the following processing stages, shown in Figure 2, finally delivering the damage prediction results:

1. Recording of the experimental sensor data (laboratory) and uploading the raw data to a file server;
2. Decoding of the raw data and storing the raw measurement data in a SQL database (numeric format with hierarchical record tables);
3. Reduction of measurement data in time and spatial dimensions (Down sampling);
4. Sensor feature selection, i.e., mapping of raw sensor data on relevant feature variables (e.g., by using FFT, discrete wavelet transformation, etc.);

5. The generation of the input data for predictor model training and inference, i.e., scaling, filtering, transformation, and reduction;
6. Supervised learning: Training and test data are needed either from experiments or simulation with sensor data associated to the appropriate target variable values (process of labelling);
7. Unsupervised learning: Only test data consists of sensor data associated with the appropriate target variable values (labelling) for model evaluation and optimization of the learning process;
8. Damage feature extraction by inference using the learned predictor model functions.

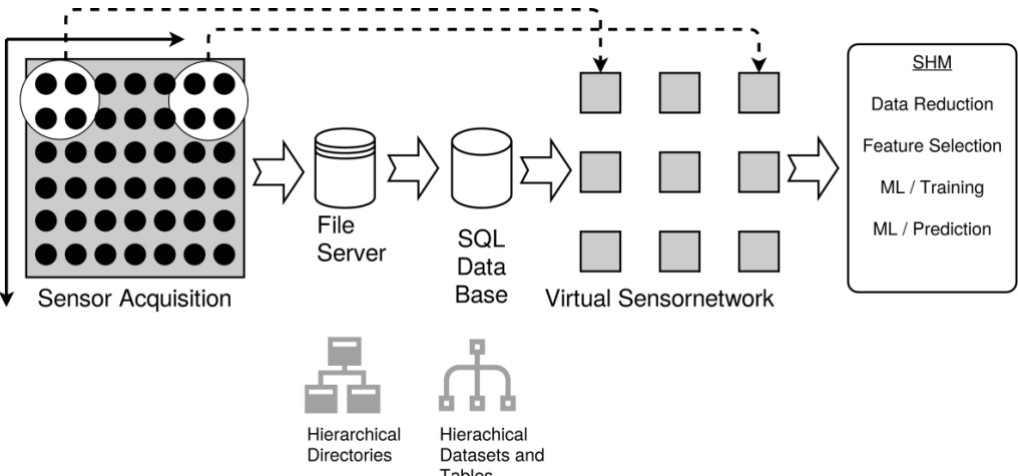

**Figure 2.** Sensor processing and data flow in the structural health monitoring (SHM) system in this work. Central part is the computation of virtual sensors arranged in a spatially two-dimensional sensor network. The dotted arrows show the relation of physical sensors and their sensor positions with virtual sensor nodes (a set of physical sensors are mapped on a virtual sensor).

Central part is an advanced SQL data base server. The SQL data base stores all experimental and computed data including ML models. The SQL data base can be accessed by a SQLJSON remote procedure call interface. The SQLJSON-RPC provides request-reply communication (e.g., SQL queries) via a JSON code and data format. SQLJSON-RPC supports micro-code execution for complex operations send by the requesting client and executed by the server. The SQLJSON-RPC APi is an overlay software layer on top of a generic SQLITE data base API. Finally, the SQLJSON service provides a virtual file system layer that maps files and directories containing data and meta data files on tables, which can be requested such as any other SQL table. To support typical data set in numerical matrix format, hierarchical data set tables were added (organising data sets similar to HDF5 structure with data types, data spaces, meta data, and automatic matrix type conversion). There is a dedicated SQLDS API with support for packed arrays on top of SQLJSON to support access of data sets (and mapping on generic SQL tables).

*2.5. Computational Complexity and Resources*

Dealing with large volumes of data is a challenge with respect to the spatial and temporal dimension. Even in case of single guided wave measurements there is a significant volume due to the time-resolved recording of the sensor signals. The spatial dimension of sensor data determines primarily storage requirements, whereas the temporal dimension determines the computational time. Computing time and storage requirements differ significantly in the different stages and phases of data processing and predictor model function training and inference:

- Phase I. Acquisition and processing of sensor data: Computational time is dominated by communication time, storage is mainly related to the original sensor data size and the communication network;
- Phase II. Preprocessing of the data (feature selection): Computational time is medium and closely related to the feature selection and transformation algorithms; storage depends still on original data size;
- Phase III. Generating of predictive models and training (with partial testing of model quality): Computational time and storage depends significantly on the used model implementation and its structure (function, directed graph/tree, neuronal networks), and computational time depends additionally on the training algorithms and the processing of the training data instances (single vs. batch vs. monolithic instance processing);
- Phase IV. Test of the trained models: Computational time depends on the model size/structure, its functional complexity, and on the number of data instances, but there is no significant increase of storage;
- Phase V. Inference/application to unknown data (incorporates Phases I/II, too): the same as Phase IV.
- Parellelisation of distributed multi-instance learning (MM or MS class) at process—or node level (basically on control path) is possible:
- All local learning instances are independent posing high computational effort that can be parallelised;
- Synchronisation and merging of local data is required only by global model fusion that is a simple task with low computational effort;
- Parallelisation can be applied on one central computer as well as in the distributed sensor network;
- Speed-up $S \leq 15$ with a central computer (2 CPU Sym. NUMA, 8 cores/CPU, L3 Cache $\geq 15$ MB);
- Speed-up $S < N$ with $N$ distributed sensor nodes (sensor data locally).

Parallelisation of single-instance learning is partially possible if there is a globally trained and generalised model that can be applied locally bounded regions of data or single sensors in the best case (SM class), i.e., a predictor functions that marks and amplifies local features from the sensor signal (posing high computational effort). Global fusion by object and pattern recognition (e.g., point density computations) or by negotiation and consent algorithms can be classified in low and mid computational classes and can be commonly neglected.

The resource and computational time requirements of both diagnosis architectures are evaluated in the respective sections.

## 3. Sensor Data and Experiments

### 3.1. Overview

An experimental data base is used to evaluate both approaches. The experiments were performed to get raw time-resolved sensor signal data $D \in \mathbb{R}^3$ featuring the following physical key facts:

- Specimen under test:
  - ○ Material: CFRP (Carbon Fibre Reinforced Plastic);
  - ○ Shape: Plate;
  - ○ Phy. dimensions: $500 \times 500$ mm;
- Measurement method: Air-coupled contactless ultrasonic 2D scan with a grid spacing of 2 mm;
- Sensor: different air-coupled ultrasonic probe;
- Measuring wave excitation by a surface bonded piezoelectric actuator;
- Excitation: Rectangular burst signal with 3 pulses at frequencies from 40 to 200 kHz;
- Defect type: Pseudo defects consisting of a round steel plate (diameter 20 mm) which is attached to the plate and provides realistic wave interactions compared to real defects;

- Defect variance: Pseudo defect boned with methyl methacrylate adhesive (MMA) or stick with vacuum sealant tape (VST) to the surface of the plate;
- Defect position variance: Experimental variations with 9/11/14 different defect positions on the plate (regularly and irregularly spaced);
  - Spatial measuring resolution (raw data): $250 \times 250$ measuring points (2 mm spacing);
  - Temporal measuring resolution: about 4000 sample points/record (amplitude).

The data was delivered in a proprietary data format and was decoded and stored in a SQL data base with hierarchical data set tables providing a close binding of measuring data and meta data (experimental parameter sets), described in Section 2.4.

### 3.2. Experimental Data Sets of Lamb Wave Propagation Fields

**Air-coupled Ultrasonic Technique**

In order to produce a high amount of experimental data sets which are required as input for the machine learning approaches the air-coupled ultrasonic technique is used. This technique is a well-established, contactless method for the measurement of Lamb wave propagation fields. To excite Lamb waves a piezoceramic actuator is applied on a plate structure. The actuator is made in form of the so-called piezocomposite technology (DuraAct, PI Ceramics GmbH) and consists of a round piezoceramic (diameter: 10 mm, thickness: 0.2 mm) to excite homogeneous, almost circular wave propagation fields. The actuator is bonded on the plate structure by a two-component epoxy adhesive in a vacuum process (Henkel AG, Loctite Hysol 9455). The transducer is driven with a rectangular burst signal with 3 pulses. The plate structure is a quasi-isotropic CFRP (Carbon Fiber Reinforced Plastic) laminate with 7 plies. The layup as well as the mechanical material properties are described in [14]. The plate dimensions are $500 \times 500 \times 2$ mm. The plate is installed on spikes inside of a frame with mechanical stops, to reduce the wave interactions with the mechanical mounting and to avoid deviation in positioning during assembly and disassembly. On the sensor side different ultrasonic sensors, which measure the out-of-plane displacement of the Lamb wave field, are used. In order to investigate the wave interaction of different Lamb wave modes ($A_0$ and $S_0$ mode) with high amplitudes (signal-to-noise ratio) the wave propagation field is measured at various frequencies (40, 80, 120, 200 kHz). At lower frequencies of 40 to 80 kHz the $A_0$ mode exhibit high amplitudes whereas the $S_0$ mode shows high amplitudes at higher frequencies of 120 and 200 kHz. The further analogue signal processing, data conversion and scanner controls are provided by the ultrasonic system USPC 4000 AirTech (Hillger NDT GmbH).

In addition, the ultrasonic system controls a portal scanner that moves the scanning sensor. The scanning sensor moves in form of a meander course over the plate and measure the wave field in a 2 mm grid. This leads to a spatial measuring resolution of $250 \times 250$ measuring points over the whole plate. The following Figure 3 shows the experimental set-up for the measurement of Lamb wave fields as well as the position of the actuator and the pseudo damages.

The output of the measurements are data files which consist of an amplitude over time signal for each measuring point within the 2D scanning grid. These so-called 3D volume data sets build the input for the following damage detection approaches. In a first step of the experimental data recording the wave propagation fields are measured at the different frequencies without any pseudo damage applied to the plate. These measurements are used as a reference for the damage detection.

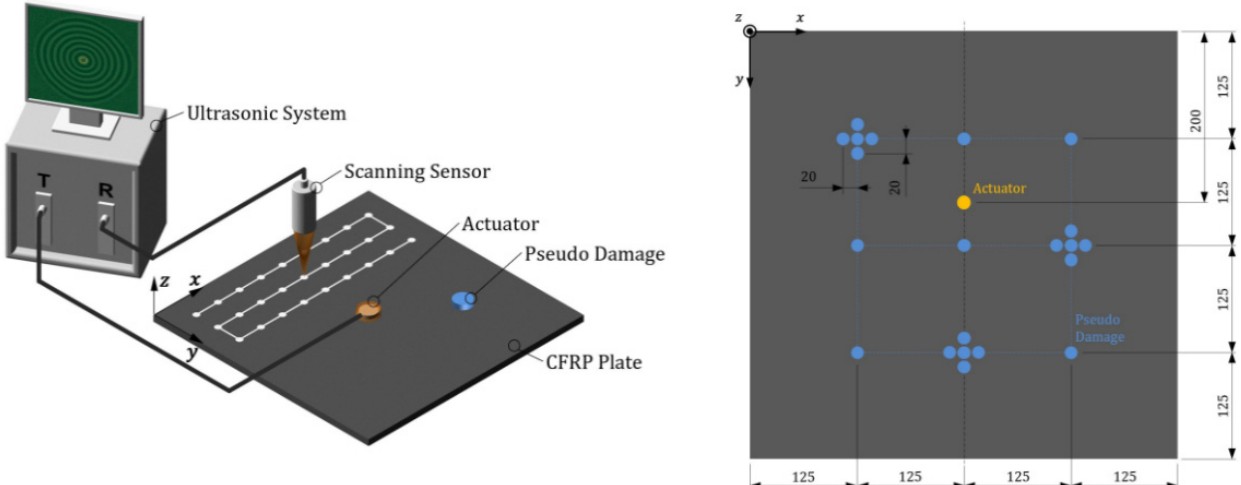

**Figure 3.** Experimental set-up for measuring Lamb wave propagation fields with an air coupled ultrasonic sensor and a scanning aperture (**left**) and position of actuator and pseudo damages (**right**), dimensions in mm.

### Removable Pseudo Damages

The machine learning approaches require a high amount of experimental data sets with different damage types and locations. To reduce the amount of manufactured CFRP plates removable pseudo damages are developed. The pseudo damages can be applied on different locations of the damage-free plate and cause comparable wave interactions such as real damages (e.g., delaminations). After each measurement set the pseudo damages can be removed without any residues. It was defined to use round pseudo damages with a diameter of 20 mm which is smaller than the required size of a damages (25 mm) to be detected by nowadays SHM systems [15]. To produce realistic wave interactions, such as absorption, reflection, scattering or mode conversion, two different types of pseudo damages are developed. The first type of pseudo damage consists of a round steel plate (thickness: 10 mm) and is applied with hand pressure to the CFRP plate using vacuum sealant tape (GS-213, Airtech Europe SARL) which will be referred in the following as VST. This pseudo damage can be removed by simply peeling it off. Due to the absorption characteristics of the relative thick sealant tape (thickness: 3 mm) this pseudo damage only absorbs the wave energy and reduces amplitudes on the sensor side.

The second type of pseudo damage consists of the same round steel plate and is bonded to the plate using methyl methacrylate adhesive (MMA). The advantage of this adhesive is that it cures within a short period of time (typical 20 min) which reduces the time span of each measurement set. Furthermore, it produces a relative thin bonding layer due to its low viscosity and exhibit high young's modulus which results in a relative rigid bonding layer and high stiffness change in the plate structure. The stiffness change causes reflections, scattering and mode conversions within the Lamb wave propagation field. This pseudo damage can be removed by a small lateral knock with a hammer. Due to the fact that the CFRP plate has a smooth surface, no residues of the adhesive are remaining on the plate. The following Figure 4 shows exemplary the Lamb wave interaction at 80 kHz with the pseudo damage applied with vacuum sealant tape.

In general, the pseudo damage applied with vacuum sealant tape exhibit an attenuation of the $A_0$ mode at 40 to 120 kHz and a phase shift at 40 to 80 kHz. Wave interaction of the S-mode with this pseudo damages are not observed in the investigated frequency range. The following Figure 5 shows exemplary the Lamb wave interaction at 80 kHz with the pseudo damage applied with methyl methacrylate adhesive.

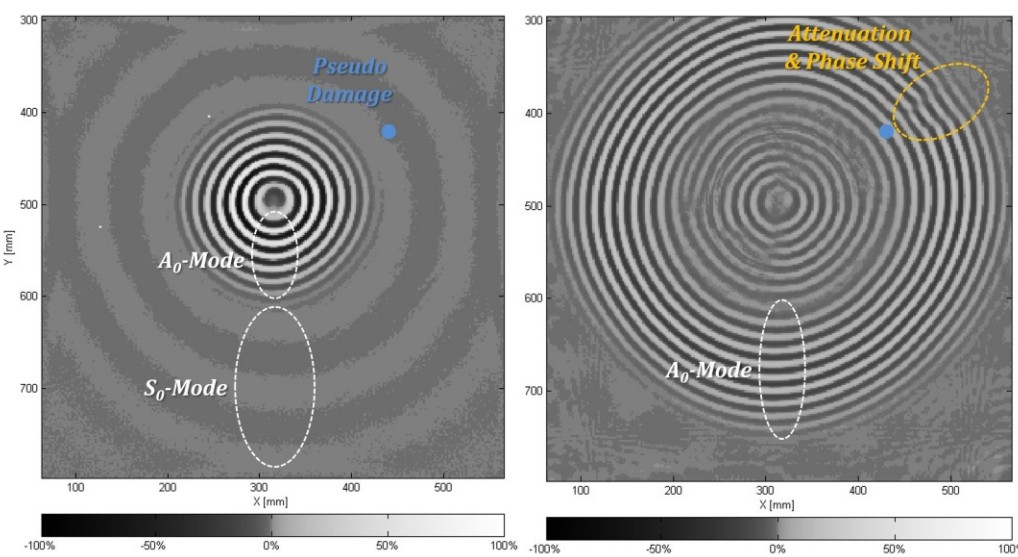

**Figure 4.** Intensity images of two-dimensional Lamb wave interaction with a pseudo damage (sealant tape) at a frequency of 80 kHz, wave propagation field at 233 µs (**left**) and at 330 µs (**right**) with highlighted region of phase shift and attenuation.

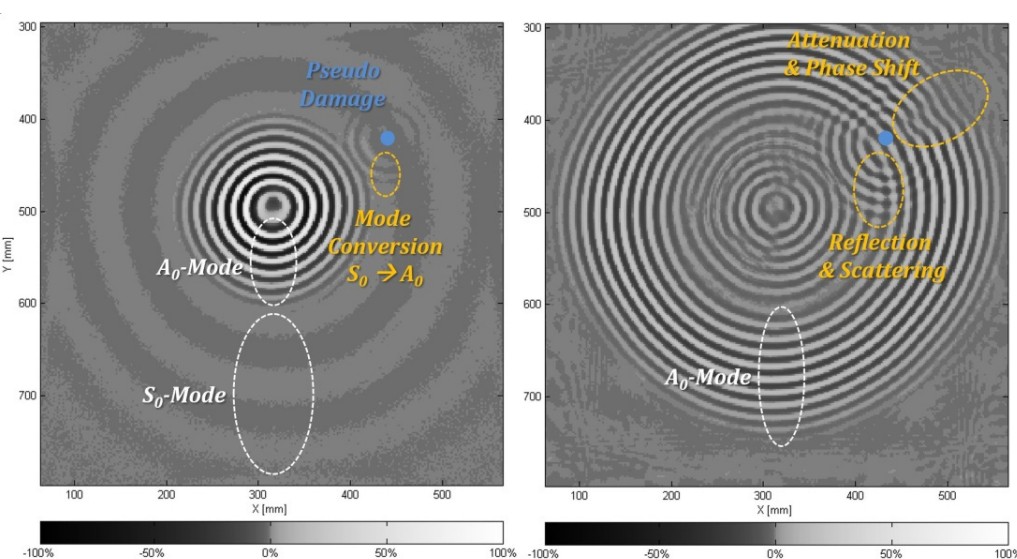

**Figure 5.** Intensity images of two-dimensional Lamb wave interaction with pseudo damage methyl methacrylate adhesive (MMA) at a frequency of 80 kHz, wave propagation field at 230 µs (**left**) and at 326 µs (**right**) with highlighted regions showing mode conversion.

This pseudo damage produces mode conversion from $S_0$ into $A_0$ mode at frequencies of 40 to 200 kHz. The $A_0$ mode exhibit reflections and scattering at 40 to 80 kHz. Wave interaction of the $A_0$ mode at higher cannot be observed because the $S_0$ mode dominates the wave propagation field with its higher amplitudes. Furthermore, local phase shifts and attenuation (behind the pseudo damage) of the $S_0$ mode from 120 to 200 kHz and of the $A_0$ mode from 40 to 80 kHz can be detected. It can be summarised, that the pseudo damage with methyl methacrylate adhesive produces more wave interactions compared to the pseudo damage with vacuum sealant tape. Therefore, this pseudo damage can be better detected by the damage detection algorithms.

The various observed wave interactions with the pseudo damages show up in the sensor signals in different ways. The mode conversion and reflection/scattering produce new wave packets which appear in the sensor signals at specific time of flights. The time of flight of the new wave packets depends on the distance between pseudo damage and sensor. Therefore, the new wave packets can be interfered by the original excited wave packets ($S_0$ and $A_0$ mode) if their time of flights matches. Or the new wave packets appear clearly in the sensor signal if their time of flights differs from the original excited wave packets. The other wave interaction, such as phase shift and attenuation, influence only the original excited wave packets in form of phase shifts and amplitude reductions. In summary, the feature selection of the machine learning algorithms should be able to identify the different wave interactions by selecting specific time frames within the sensor signals. Within the experimental data sets the two types of pseudo damages are applied one after the other at 21 different positions, as shown in Figure 3. At each position and with each pseudo defect the wave propagation field is recorded at the all defined frequencies (40, 80, 120, 200 kHz).

### 3.3. Signal Features and Damage-Wave Interaction

As outlined in the previous section, damages or more general material inhomogeneity have an influence on the wave propagation with respect to:

- Amplitude modification, i.e., damping and inference;
- Reflection;
- Frequency and mode conversion.

Therefore, relevant features of the measured temporally and spatially resolved sensor signals are related to amplitude, phase, and frequency properties. But the time-resolved sensor signal at a specific measuring position will consist of different segments. Typically, only the first segments contain damage-relevant features, whereas the later segments

There are basically three different approaches for extracting relevant signal features (beside statistical properties):

- Time-frequency transformation (Fourier transform) of the entire signal record;
- Time-shifted window frequency transformation;
- Wavelet transform and decomposition of the signal record [16,17].

The frequency transformations are bound to the time-frequency uncertainty principle, and a windowing approach increases the time resolution but decreases the frequency resolution. The wavelet transform (among other wave decomposition methods not discussed here) can be considered as method preserving time and frequency properties of the input signal [16].

The relevant damage features are contained in the sensor signal. Bit temporal position and extend of the region of interest containing the relevant features depends on the wave propagation, the wave interaction, and the relative positions of the sensor, damage, and actuator (i.e., forming a spatial graph), shown in Figure 6b. The sensor signal is basically divided into three segments of initially unknown length: The pre-, feature, and post-signal segments, illustrated in Figure 6a. Even the comparison with a base-line (damage-free) signal does not expose the relevant features without pre- and post-processing.

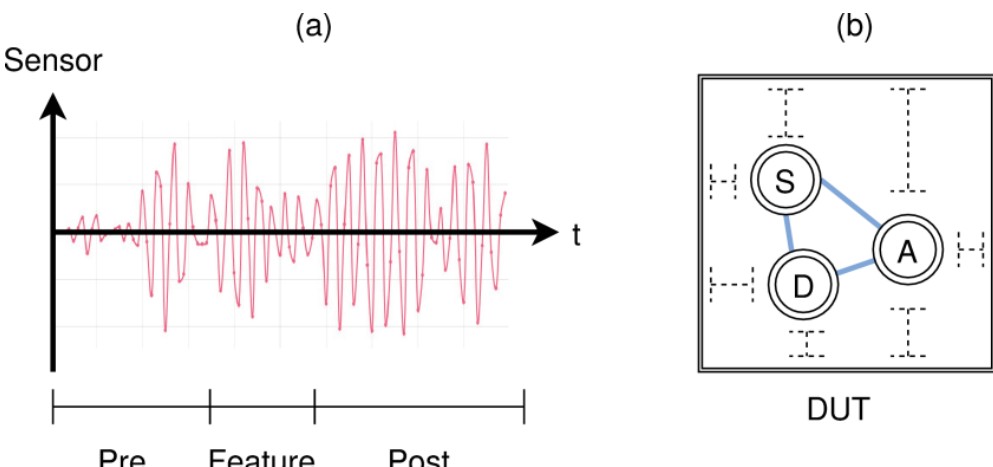

**Figure 6.** (**a**) The time-resolved sensor signal (amplitude) can be divided into three segments with only the middle segment containing relevant damage features. (**b**) Spatial dependencies of the spatial sensor-damage-actuator configuration with effect on wave propagation and damage detection.

## 4. Multi-Instance Learning with Multi-Instance Prediction (MTMP Class)

In this section the first low-resolution and low-resource approach using multi-instance learning of a damage predictor function is introduced. In a first attempt, the raw sensor data is processed by a virtual sensor network on a generic computer. The results can be mapped directly on a real sensor network.

### 4.1. Concept

The damage diagnostics processing the raw sensor data uses the following key methods:

- Supervised multi-instance learning by a virtual sensor network ($8 \times 8$ nodes) processing local time-resolved sensor data derived from the experimental measuring data;
- The output of the local supervised learning is a predictor function that have to detect a damage in the near region around a sensor (continuous output in the range [0,1] with binary threshold classification);
- The predictor function is implemented by a state-based recurrent ANN with LSTM cells by using a JavaScript Ml framework integrating an improved Neataptic ANN [18], the network configuration is [1,4,6], i.e., 4 input neurons, 6 LSTM cells, and one output neuron;
- The global fusion of all local damage predictor function outputs approximates the spatial damage position (if any) within the boundary of the sensor network supporting position interpolation.

The principle experimental and data analysis set-up is shown in Figure 7. All computations were performed in JavaScript either by a WEB browser (SpiderMonkey VM) or by using node.js (V8 VM).

### 4.2. Feature Selection and Network Architecture

1. Down sampling of the raw sensor signal (approx. 4000 Samples on the time axis) with a 1:10 ratio (sampling every n-th sample)
2. Spatial data reduction at the location of a virtual sensor $s(x,y)$ by using a $2 \times 2$ pixel field that is reduced to one virtual sensor (spatial mean averaging)
3. Temporal down sampling of the reduced sensor signal 1:4 (temporal mean averaging)
4. Discrete wavelet transform (DWT with Debuchet-1 function) LP/HP filters (4 levels with Up-Sampling) [16,17,19].

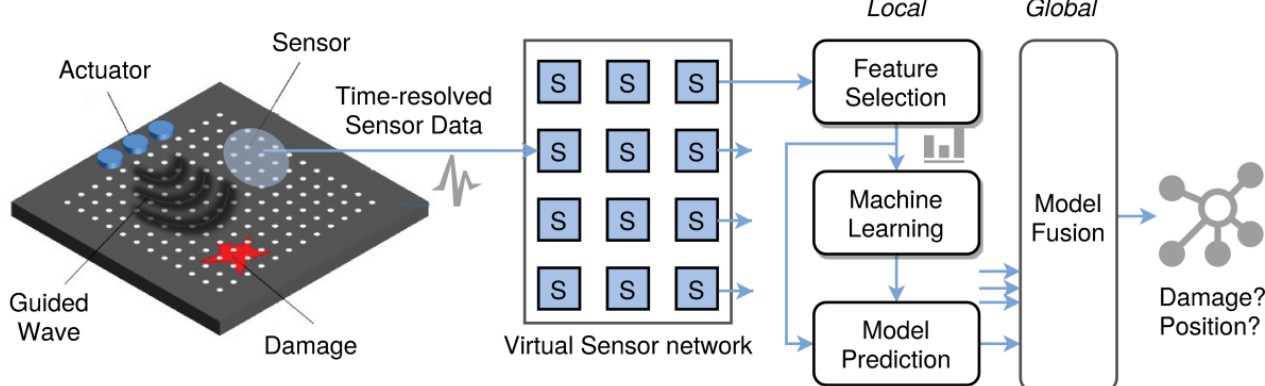

**Figure 7.** Measuring and data processing architecture computing a virtual sensor network with $8 \times 8$ nodes and a spatial sensor distance of 60 mm from the original measured physical sensor data.

The feature selection process and the basic ANN architecture is shown in Figure 8. Typically, the levels 3–5 contain relevant signal features. Each level of the DWT consists of a low- and high-pass filter providing the approximation and details of the signal, respectively, providing a good time-frequency analysis [17]. The approximation is the input signal of the following next level, the detail signal is the input for the ANN. Each level of the DWT reduces the sampling frequency by two (down sampling), i.e., at the output of the DWT filter a sampling expander (up sampling) is required for each level to equalise the sequence length of each input signal.

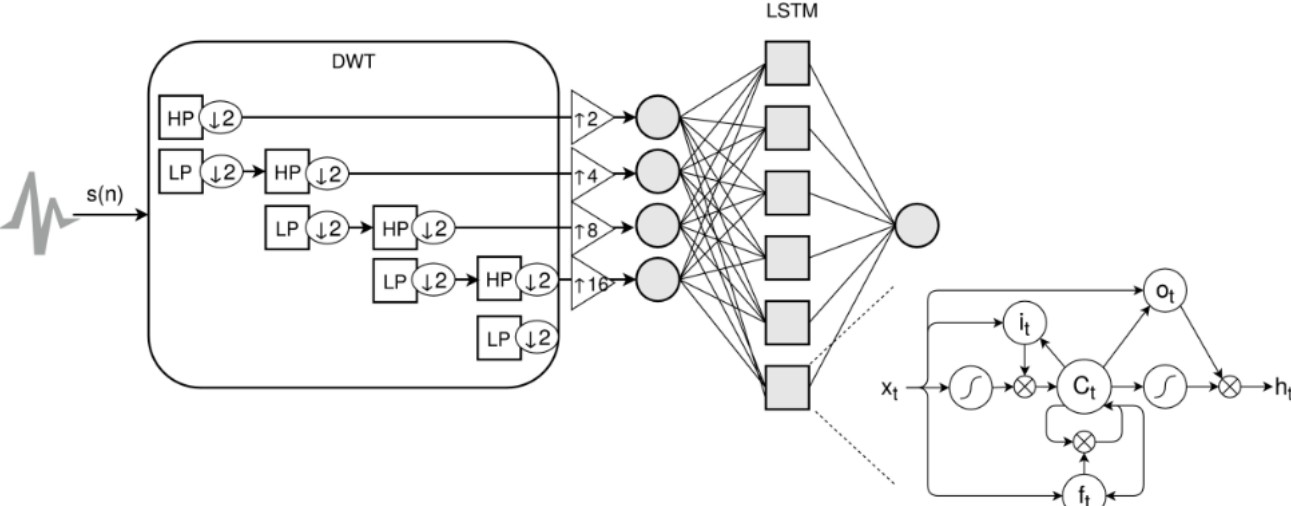

**Figure 8.** Feature selection with discrete wavelet transform (DWT) decomposition from time-resolved and discretised sensor signal $s(n)$ as the input for the damage predictor function implemented by a recurrent artificial neural networks (ANN) using long-short term memory (LSTM) cells (LSTM cell drawing from [18]). HP: high-pass filter; LP: low-pass filter; ↓: down sampling; ↑: up sampling; $o_t$: output fate, $f_t$: forget gate; $i_t$: input gate; $C_t$: memory cell.

The recurrent state-based ANN structure consists of $n$ input neurons with sigmoid transfer function (one for each DWT decomposition level used), a hidden layer of Long-short term memory cells (LSTM), and one output neuron with an output range of [0,1] (also sigmoid transfer function). A value nearby 1 represents the detection of a damage in the surrounding region around a sensor.

There are many different LSTM cell architecture around. We are using the LSTM cell implementation from the Neataptic ML framework [18], shown on the right side of Figure 8. Central part is a state cell ($C_t$) surrounded by different gates (input $i_t$ and output $o_t$ gates) controlling the forward and feedback paths of the cell and the memory history (by

the forget gate $f_t$). Depending on the particular configuration, the LSTM cells of one layer can be interconnected (memory-to-memory connections).

The DWT for the i-th level can be generally defined by the detail and approximation functions $D$ and $A$ related to the high- and low-pass filters, respectively:

$$D_n = \frac{1}{\sqrt{N}} \sum_{i=0,...,N-1} x(i) \times \psi_{j,k}(i)$$
$$A_n = \frac{1}{\sqrt{N}} \sum_{i=0,...,N-1} x(i) \times \phi_{j,k}(i), \tag{4}$$

with $N$ data points of the original time series $x(i)$, $i = 0, 1, \ldots, N-1, j = 0, 1, \ldots, J-1, k = 0, 1, \ldots, 2^J-1, J = \log(N)$. The function $\psi$ and $\varphi$ are related to the mother wavelet function and its mirror function, respectively.

Details regarding the DWT can be found in [17].

### 4.3. Target Variable Computation for Labelling

The individual sensor nodes should detect damage/defects within a local area. There is a simplified assumption that damage detection is possible in a circular area around a sensor, i.e., isotropic sensitivity (not true; rather elliptically shaped in direction of the axis actuator-damage-sensor). A Euclidean distance damage-sensor is used as an indicator of damage/non-damage classification, i.e., a specification for the expected prediction value of the ML model.

$$y_{i,j} = \begin{cases} \sqrt{(dam_x - p_x)^2 + (dam_y.p_y)^2}) < R, 1 \\ else, 0 \end{cases}, \tag{5}$$

with $p$ as the sensor position. The target variable estimation is only required for the first supervised learning approach. The second unsupervised approach due not rely on labelling for training, shown in Figure 9.

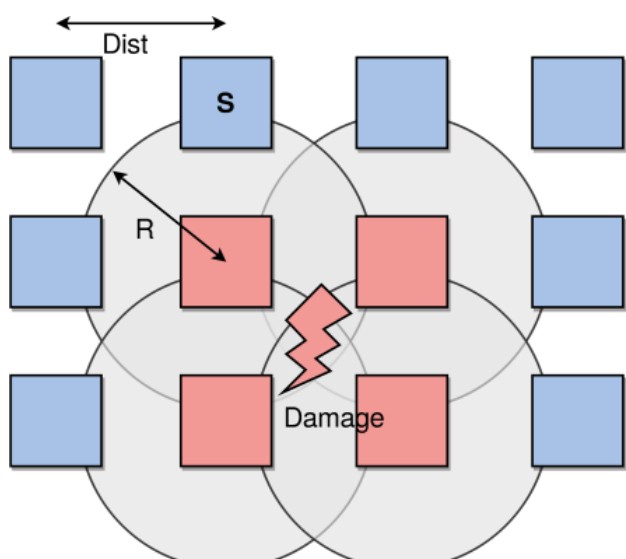

**Figure 9.** Labelling of training data (assigning target output variable outcomes) by simple neighbourhood detection of damages in the range $2R$ around a sensor node centre position. Shown is a part of the sensor network (red colour: damage within radius $R$, blue colour: no damage).

### 4.4. Global State Fusion and Damage Localisation

Each node delivers a probalistic damage estimation within a radius $R$ around each node. The local predictor function delivers an output value in the range [0,1]. Values near one indicate strong damage detection. The output of all local predictor functions are fused to a global state (binary damage classification and damage localisation relative to

the DUT coordinates) by different algorithms described below. Typically clusters of nodes are activated around the damage location. Beside true-positive predictions, there can be false-positive predictions, too. The global fusion has to discriminate the false-positive predictions and averages the local states.

Global fusion algorithms used for damage localisation:

**Unweighted Centre of Mass.**

This algorithm applies a threshold filter to all local prediction results with a binary decision mapping (damage activation). All activated discrete node positions are added to a point cloud. Finally, an unweighted centre of mass (COM) computation is applied to this point cloud interpolating the damage position $p_d$.

$$p_d(SN) = \left( \sum_{s \in SN} \left\{ \begin{array}{l} 1, s_o > t \\ 0, else \end{array} \right\} \right)^{-1} \sum_{s \in SN} \left( \begin{array}{c} s_x \\ s_y \end{array} \right) \left\{ \begin{array}{l} 1, s_o > t \\ 0, else \end{array} \right\}, \tag{6}$$

where $SN$ is the full set of sensor nodes of the network, $s_o$ the output of the prediction function of the node at position $s_x, s_y$ in the range [0,1], $t$ is the threshold for binary classification.

**Weighted Centre of Mass.**

This algorithm applies a threshold filter to all local prediction results with a binary decision mapping (damage activation). All activated discrete node positions are added to a point cloud together with a weight derived from the predictor function output. Finally, a weighted centre of mass computation is applied to this point cloud interpolating the damage position.

$$p_d(SN) = \left( \sum_{s \in SN} \left\{ \begin{array}{l} s_o, s_{pred} > t \\ 0, else \end{array} \right\} \right)^{-1} \sum_{s \in SN} \left( \begin{array}{c} s_x \\ s_y \end{array} \right) \left\{ \begin{array}{l} s_o, s_o > t \\ 0, else \end{array} \right\} \tag{7}$$

**Fully weighted Center of Mass.**

This algorithm creates a point cloud with all discrete node positions together with the weight derived from the predictor function output. Finally, a weighted centre of mass computation is applied to this point cloud interpolating the damage position.

$$p_d(SN) = \left( \sum_{s \in SN} s_{pred} \right)^{-1} \sum_{s \in SN} \left( \begin{array}{c} s_x \\ s_y \end{array} \right) s_o \tag{8}$$

**Density-based Clustering and Center of Mass.**

Prior a weighted centre of mass computation, a density-based clustering using the DBSCAN algorithm [20] is applied to the point cloud consisting of node positions with a predictor function output above a given threshold. The largest clustered group is select for COM. This approach is proposed to be useful to discriminate clusters of true-positive predictions from clusters of false-positive predictions, as evaluated and discussed in the following results section. The DBSCAN algorithm uses a global density parameter. An advanced approach can be used with a local density parameter uses for clustering [21].

**Distributed Center of Mass (Cellular Automata).**

The previous algorithms collect all node prediction results and perform the damage localisation on a dedicated centralised node. To avoid any centralised instances for scalability and robustness reasons a distributed COM algorithm is processed by the network nodes, i.e., an algorithm based on a cellular automata model with neighbourhood communication only. The algorithm bases on the fully weighted COM approach and Cellular Automata (CA).

The basic concept of a distributed weighted COM (DCOM) is the propagation of partials sums in rows and columns of the network, assuming more or less logical regular grid communication architecture. The logical position of a node with respect to the sensor network have to satisfy an ordering constraint, i.e., an East neighbour is physically located on the right, a West neighbour is located on the left side, and so on.

The first upper left node of the CA network initiate the propagation of the partial sum calculation from left to right (horizontal axis) and downwards (only initiators of further row propagations). Each node at the end of the row propagates the row accumulation downward. The last lower right node finally computes the approximated centre position of the damage. Each cell has a state, defined in Algorithm 1. Only the first node must be marked (always position (1,1)). All other nodes derive their position from the neighbouring nodes, i.e., a node has not to know its absolute position in the network, only the relative neighbouring connectivity.

Assuming a regular mesh sensor network with $N \times M$ nodes the DCOM approach requires $NM$ steps to compute the weighted damage position.

---

**Algorithm 1. Data structure of CA cell**

---

```
1:  type cell = {
2:      state : {
3:          activation : number,
4:          prediction : number,
5:          // accumulator for horizontal axis propagation
6:          right : comsum {} | null,
7:          // accumulator for vertical axis propagation
8:          down : comsum {} | null,
9:          // accumulator for accumulated row results
10:         row : comsum {} | null,
11:         position : [x,y],
12:      },
13:      activity : function
14: }
15: type comsum = {
16:     x: number, y:number,
17:     accux: number, accuy:number,
18:     mass: number,
19: }
```

---

The cell activity is shown in the Algorithm 2, and the principle right-down shift propagation of the weighted COM is shown in Figure 10. There is a dedicated initiator and collector node. Note that any edge node can be initiator or collector by rotation of the node matrix. All four configurations can be processed overlapped increasing redundancy in a technical network with node or communication failures.

---

**Algorithm 2. CA cell COM accumulation algorithm**

---

```
1:  cell.activity = function (neighbors,x,y) {
2:      if !neighbors.left and not neighbors.up) then
3:          if !neighbors.right and not neighbors.down then
4:              // Initiator
5:              right := {mass:prediction,x:1,y:1,
6:                          accux:prediction,accuy:prediction}
7:              down := {mass:prediction,x:1,y:1,
8:                          accux:prediction,accuy:prediction}
9:              position := [1,1]
10:         else
11:             if neighbors.right and neighbors.left and neighbors.left.right then
12:             right := copy(neighbors.left.right)
13:             right.mass :=+ prediction
14:             right.x :=+ 1
```

---

| **Algorithm 2. CA cell COM accumulation algorithm** |
| --- |
| *15:*          right.accux :=+ (prediction*right.x) |
| *16:*          right.accuy :=+ (prediction*right.y) |
| *17:*          position := [right.x,right.y] |
| *18:*     **if** !neighbors.left and neighbors.up and neighbors.up.down **then** |
| *19:*          *// Next row initiator* |
| *20:*          right := {mass:prediction,x:1,y:neighbors.up.down.y+1, |
| *21:*                accux:prediction,accuy:prediction} |
| *22:*          position := [right.x,right.y] |
| *23:*          **if** neighbors.down **then** |
| *24:*             down := {mass:prediction,x:1,y:up.down.y+1, |
| *25:*                accux:prediction,accuy:prediction} |
| *26:*     **if** !neighbors.right and neighbors.left.right **then** |
| *27:*          *// horizontal end point* |
| *28:*          *// Propagate* |
| *29:*          **if** !neighbors.up **then** |
| *30:*             row := copy(neighbors.left.right) |
| *31:*             row.x :=+ 1 |
| *32:*          **elseif** neighbors.up and neighbors.up.row **then** |
| *33:*             *// accumulate with previous row* |
| *34:*             row := copy(left.right) |
| *35:*             row.x :=+ 1 |
| *36:*             row.mass :=+ prediction |
| *37:*             row.accux :=+ (prediction*row.x) |
| *38:*             row.accuy :=+ (prediction*row.y) |
| *39:*             row.mass :=+ up.row.mass |
| *40:*             row.accux :=+ up.row.accux |
| *41:*             row.accuy :=+ up.row.accuy |
| *42:*          position := [row.x,row.y] |
| *43:*          **if** !neighbors.right and not neighbors.down **then** |
| *44:*             *// final edge computation* |
| *45:*             center := [row.accux/row.mass,row.accuy/row.mass] |

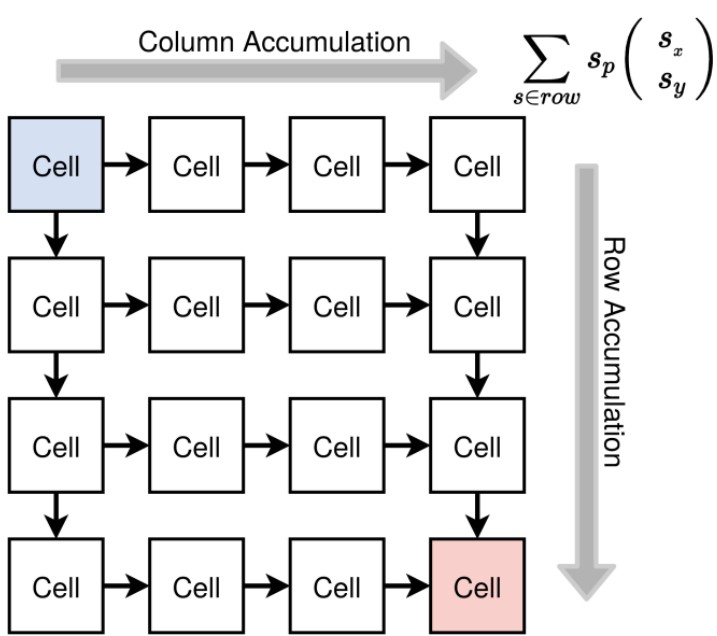

**Figure 10.** Shift operations of data in the distributed CA-based COM. The blue node is the initiator, and the red node is the collector. Any edge node can be initiator or collector by rotation of the node matrix by 90 degree.

*4.5. Training*

The training of the RNN is rather simple. All model instances associated with virtual sensor nodes are trained independently (although, on a central computer sequentially). The feature transformed input signals activate each RNN sequentially. The first four DWT decomposition levels are used. After the RNN is activated, the prediction error is computed. It is just the difference of the target variable (binary damage label) at the last output value of the RNN (linearised in the interval [0,1], 1: damage, 0: no damage)). The desired target variable value (0/1) is passed to a gradient descent back propagation algorithm adapting the weights of the network and the parameterisation of the LSTM cells (primarily internal edge weights and gating parameters).

The basic training algorithm for one node is shown in Algorithm 3.

---

**Algorithm 3. Basic training algorithm for one node and one training sample**

---

*1:*  **function** train(node,data) **is**
*2:*  　target=data.target
*3:*  　error=0
*4:*  　**repeat** *n* times
*5:*  　　node.model.clear()
*6:*  　　$\forall$ val $\in$ data,signal **do**
*7:*  　　　out=node.model.activate(val)
*8:*  　　node.model.propagate(*rate,momentum,*[target])
*9:*  　　error := error+ |out-target|
*10:*  　error := error/*n*
*11:*  　**if** error>0.2 **then**
*12:*  　　**if** target>0.2 **then** error0 := 0, error1 := 1
*13:*  　　**else** error0 := 1, error1 := 0
*14:*  　**else** error0 := error1 := 0
*15:*  　**if** target>0.2 **then** state := 1 **else** state := 0
*16:*  　**return** error,error0,error1,state
*17:*
*18:*  **function** test(node,data) **is**
*19:*  　target := data.target
*20:*  　error := 0
*21:*  　node.model.clear()
*22:*  　$\forall$ val $\in$ data,signal **do**
*23:*  　　out := node.model.activate(val)
*24:*  　error :=+ |out-target|
*25:*  　**if** error>0.2 **then**
*26:*  　　**if** target>0.2 **then** error0 := 0, error1 := 1
*27:*  　　**else** error0 := 1, error1 := 0
*28:*  　**else** error0 := error1 := 0
*29:*  　**if** target>0.2 **then** state := 1 **else** state := 0
*30:*  　**return** error,error0,error1,state

---

The training is applied to all nodes with a randomly sequential selection of training instances. After a spatially averaged mean error is below a threshold value, selected nodes with false-positive and/or false-negative predictions are trained, shown in Algorithm 4. The false-positive rate for the non-damage case must be zero, the local false-positive or false-negative rates in damage experiments should be minimised.

**Algorithm 4. Iterative and adaptive training algorithm for all nodes and all samples**

```
1:   // Average Training
2:   phase 1:
3:     while errorT0 > thres0 and errorT1 > thres1 do
4:        sample := random.select(trainingData);
5:        ∀ node ∈ nodes do
6:          {error,error0,error1,state} := train(node,sample)∪
7:          errorT0 := 0.9*errorT0+0.1*error0;
8:          errorT1 := 0.9*errorT1+0.1*error1;
9:
10:  // Cross Validate
11:  phase 2:
12:        maybenodes=[[]],badnodes=[[]];
13:        ∀ node ∈ nodes do
14:          ∀ sample ∈ testData ∪ trainingData do
15:            {error,error0,error1,state} := train(node,sample)
16:            if error0 > 9.5 and node ∉ badnodes then
17:              add node to badnodes
18:              if node ∈ maybenodes then
19:                remove node from maybenodes
20:            else if error > 0.5 and node ∉ badnodes ∪ maybenodes then
21:              add node to maybenodes
22:  // Selected Training
23:  phase 3:
24:        while badnodes not empty do
25:          ∀ node ∈ badnodes do
26:            phase1 with nodes=[node]
27:            phase2 wtih nodes=[node]
```

### 4.6. Results

In Figure 11, some results of the distributed sensor network activation and damage prediction are shown for the training set consisting of 9 damage positions (MMA) and one base-line experiment.

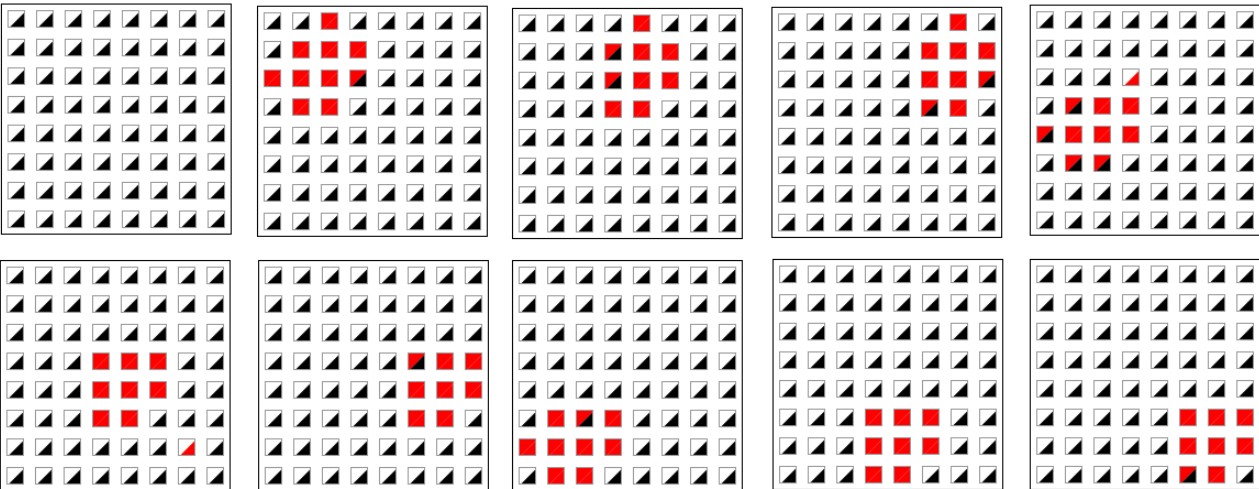

**Figure 11.** Examples of local predictor function activations (red colour, binary categorisation by a threshold function) for each sensor node for one base-line experiment and 9 experiments with different damage positions. Each square shows in the upper triangle the damage label marking and in the lower triangle the actual damage prediction from sensor data.

The following Table 1 and the bar plot in Figure 12 shows the prediction accuracy of the trained LSTM model using DWT features of the time-resolved sensor signal. The positions errors of the weighted centre point calculation of a predicted damage (pseudo defect) is in mm and must be evaluated with respect to the overall DUT plate dimension of 500 × 500 mm and the sensor node spacing of 60 mm. The prediction accuracy is averaged over all data sets. The first data set was used for the ANN training and for the test evaluation. The mean position accuracy is about 60 mm averaged over all experiments and data sets, i.e., in the order of the sensor node spacing distance (60 mm). The mean position accuracy is about 20 mm for training data experiments only, i.e., 1/3 of the spatial sensor node spacing distance.

**Table 1.** All US signal measurements with 80 kHz signal frequency, Monte Carlo Simulation (MCS) with 10% multiplicative Gaussian noise augmentation and data augmentation (10 augmented measurements for each data instance), fully weighted centre of mass computation used to estimate damage position.

| Series | Description | Mean Position Error | Min/Max Error | Std. Deviation | False Rates |
|---|---|---|---|---|---|
| 1 | Training and test set, pseudo defect MMA mounting, nine defect positions and base-line | 23 mm | 3/64 mm | 14 mm | 0/0 FP/FN |
| 2 | Test set, pseudo defect VST mounting, nine defect positions | 60 mm | 5/129 mm | 30 mm | 0/0 FP/FN |
| 3 | Test set, mixed pseudo defect mounting (VST/MMA), 21 positions and base-line | 65 mm | 18/158 mm | 32 mm | 0/0 FP/FN |

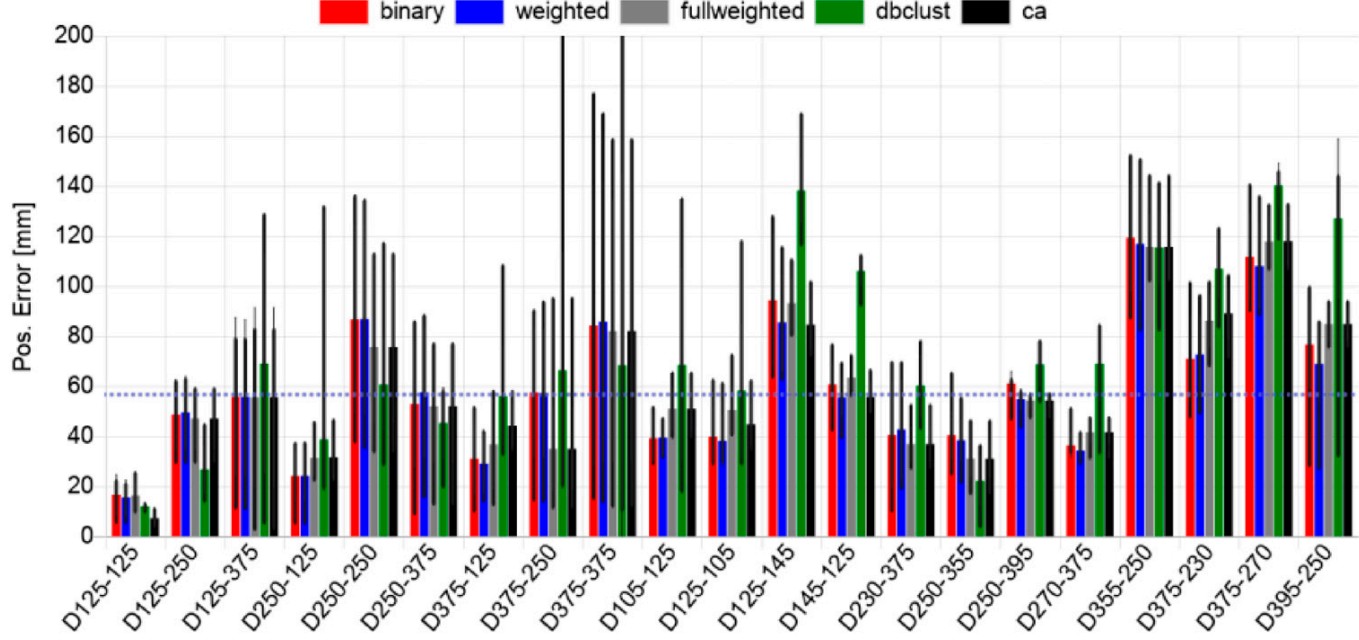

**Figure 12.** Accumulated prediction error statistics for all defect positions and all data series (21 pseudo defect positions and two mounting technologies vacuum sealant tape VST/MMA) with MC simulation adding 10% multiplicative Gaussian sensor noise (Label is defect position D⟨px⟩-⟨py⟩ in mm). Five different damage localisation algorithms are compared applied to local prediction results. The grey lines indicate the standard deviation interval 2σ and minimal and maximal errors in a set.

In Figure 12, five different global fusion algorithms are compared (see Section 4.4 for details). In most damage cases the fully weighted COM approach shows the best average accuracy results. Some damage cases show still good average position accuracy but with larger variance and in few cases with a large maximal error (e.g., D375-250),

i.e., extending the error boundary, another important statistical feature of a SHM system. This shows the dependence of the damage position estimation from the spatial sensor-actuator-damage triangle and their positions relative to each other and relative to the edges and sides of the plate. At the edges there are significant wave distortion effects, such as edge reflections, with a significant impact on the damage prediction. Fortunately, due to the spatial specialisation of the trained predictor functions of the sensor nodes near the edges and sides of the plate they are able to discriminate these wave distortions sufficiently.

In Figure 13 some typical network activation patterns with local false-positive activation clusters are shown. The density-based clustering approach can lower the average of the damage localisation error, but increases the maximal error boundary. This increase of the maximal error is a result of (1) Wrong cluster discrimination (selecting the cluster with the highest number of points), and (2) the false-positive prediction compensate a position estimation by a geometrical distorted (non-separable) true-positive cluster. The binary unweighted COM approach using a threshold discrimination produces only in some cases lower localisation errors. The fully weighted COM approach shows mostly the best results. The distributed approach with a CA model shows comparable results and is fully suitable to approximate the damage position.

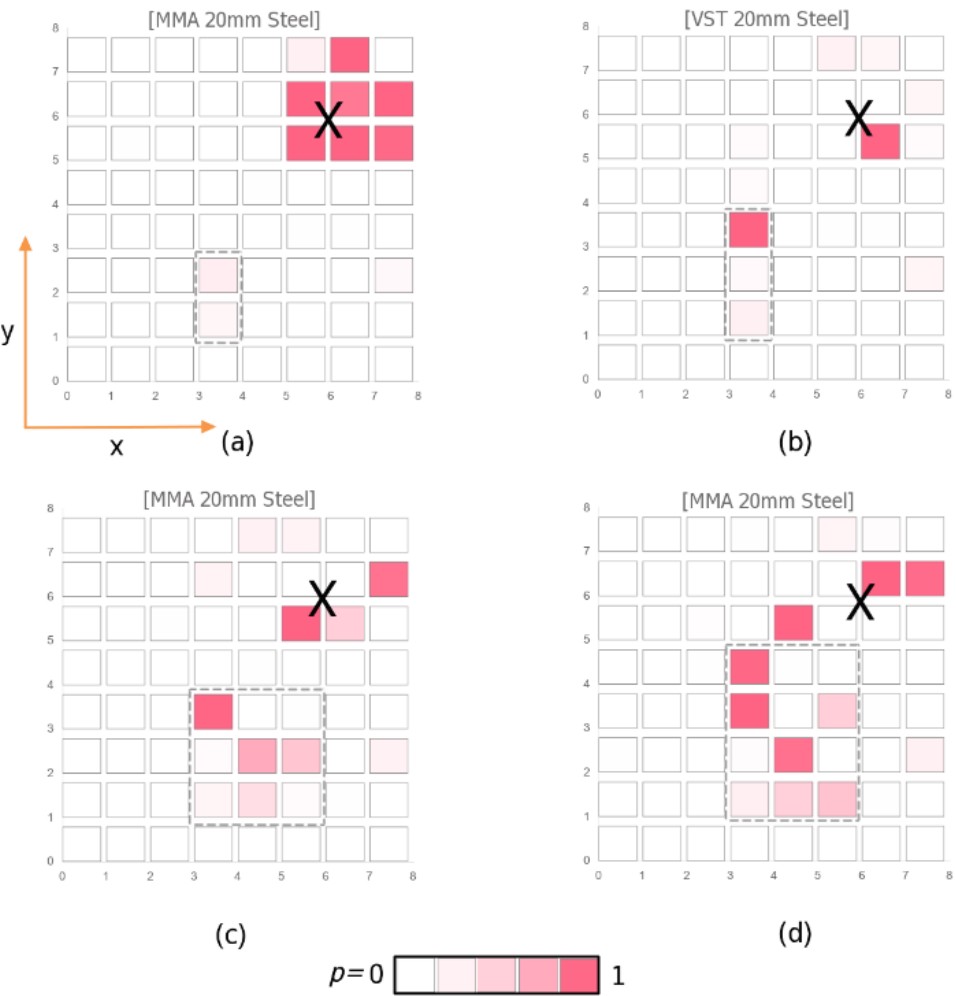

**Figure 13.** Examples of sensor network activation patterns with local false-positive activations (X: Damage position, grey dotted rectangle: cluster of false-positive activations). Shown is the raw numerical output in the range [0,1] of the two-dimensional prediction matrix of the sensor network with 8 × 8 nodes; (**a**) One strong cluster; (**b**) Two weak clusters; (**c**) Two separated clusters; (**d**) Two nearby extended clusters.

## 5. Single-Instance Learning of an Auto-Encoder with Multi-Instance Prediction (STMP Class)

In this section, the second high-resolution approach using unsupervised generalised single-instance learning of a signal auto-encoder is introduced. The output of the trained auto-encoder is used to predict the damage position (pseudo defect) by using weighted point density (WPD) analysis.

### 5.1. Concept

In contrast of the concept of the MTMP approach directly predicting damage features, the second STMP approach consists of two stages:

1.  A anomaly feature marking by a RNN detecting difference of the sensor data to a non-damage base-line experiment;
2.  A damage feature extraction using the output from 1.

A predictor function is trained using data only from a damage-free baseline experiment. Any non-conformity to the base-line data (features) is detected by the predictor function with a "damage" classification. The challenge is to derive a generalised predictor function (independence from spatial location of sensor, actuator, and damage) which discriminates damages from other signal non-conformity, i.e., noise, variance in the measuring configuration, reflection of waves at edges, and many more non-damage related artifacts. It can be assumed that there are commonly sufficient training data sets with varying damage-free sample instances, i.e., with a variance in operational and measuring conditions.

One unsupervised method to detect differences to a base-line signal is using an auto-encoder and decoder to code and reconstruct (decode) the sensor signal. If the auto-encoder function is trained only with damage-free sensor signals it is not able to reconstruct a signal resulting from wave interactions nearby a damage. Comparing the reconstructed signal with the actually measured signal gives a binary damage classificator by applying a threshold function to the mean average error of the reconstructed signal and the originally measured signal. The basic signal processing architecture is shown in Figure 14.

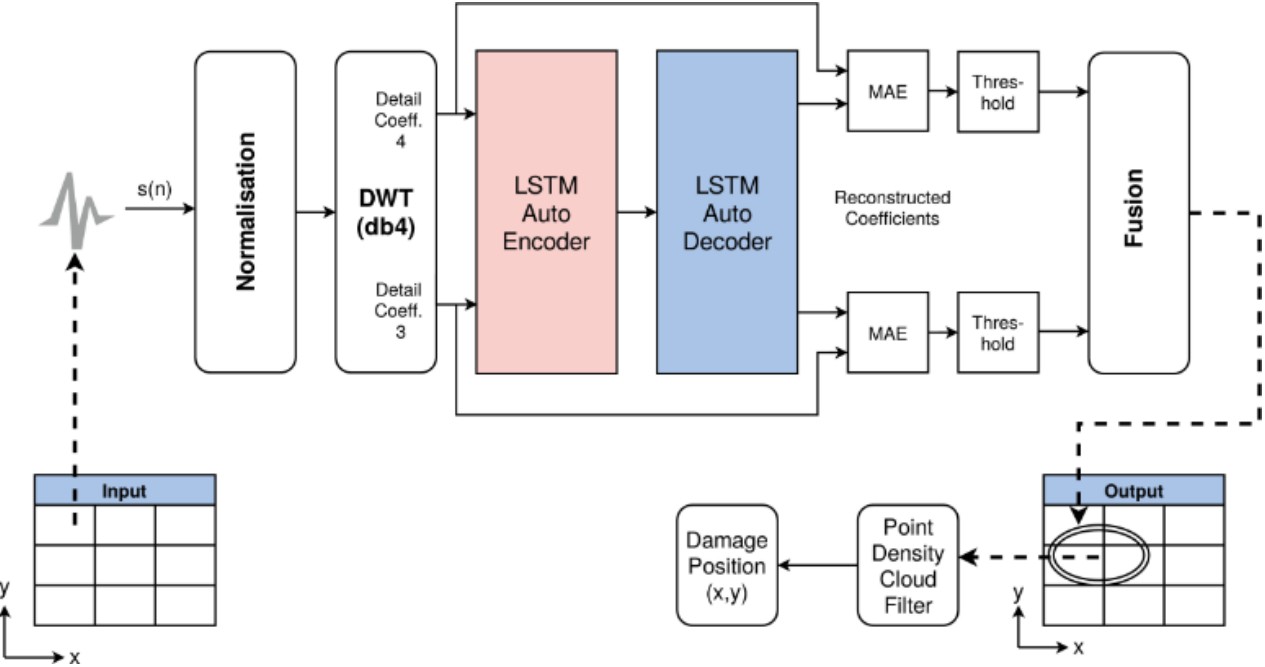

**Figure 14.** Basic architecture of the auto-encoder-based damage predictor function (MAE: Mean Average Error). The first stage normalises the sensor data and performs down-sampling. This pre-processed data is passed to the DWT calculator, and its output is the input of the LSTM auto-encoder and decoder. Finally, a fusion of the MAE values is performed.

The auto-encoder and decoder was implemented with a state-based recurrent ANN and LSTM cells using the Tensorflow ML framework [22]. Computation (training and inference) was primarily performed on a GPU. The network configuration is [64,32,32,64,2], i.e., an input layer of 64 LSTM cells, a hidden layer of 32 LSTM cells (encoder) and a hidden layer of 64 and 32 LSTM cells implementing the decoder, and two output neurons.

### 5.2. Feature Selection and Training

Similar to the MTMP class with supervised training, the single model instance is trained unsupervised with features selected from the sensor signal, processing the sensor signal in the following order:

1. Temporal down sampling 1:10 going from 4000 samples down to 400;
2. Transform the data by scaling the input between −1 and 1;
3. Discrete Wavelet Transformation (Debuchet-4 function) with 4 levels of decomposition;
4. Third and fourth down sampled level coefficients serve as the input features for the network.

Note that there is no data labelling performed. The single model instance is trained with all spatial data points (250 × 250) sequentially from the original scanned ultrasonic measurements. The training is performed again with a network activation of the down-sampled and DWT feature transformed input data sequence, with a following back propagation of the prediction error. In this case the error is defined by the difference between the input signal and the reconstructed output signal by the auto-encoder and decoder. Details of the network architecture can be found in the next section.

### 5.3. Network Architecture

The network is based on LSTM-cells arranged in an encoder-decoder setting, shown in Figure 15. Both the encoder and the decoder consist of 2 layers of LSTM-cells with a decreasing/increasing amount of units respectively. This arrangement serves as a bottleneck where only the most essential information from the input features are kept. The compressed information is then used to decode it back to its original form. The network therefore is an LSTM-based auto-encoder. This also means that prior labelling becomes unnecessary as it is an unsupervised learning technique. By training the network with global data of undamaged CFK-plates, it learns to accurately compress and decompress its undamaged input data on any local position individually. However, supplying the network with signal data that includes damage information, e.g., wave reflections, results in a much greater error, because the network intentionally never saw damage information during training. The reconstruction error of the network is therefore an indication of a possible defect.

### 5.4. Post Processing

The mean averaged error derived from the decoder output is then classified into damage or no damage features using a simple threshold function. Because of the globally trained network this procedure can then be repeated for sensors with different locations on the CFK-plate, which, applied iteratively, results in a binary image of spatially resolved damage/no damage feature classifications of any resolution. This image can then be used as the input for a weighted point density analysis using DBSCAN to estimate the damage location.

Typical examples of the post-processed images are shown in Figure 16. The dependency of the position accuracy with respect to the sensor-actuator-damage configuration is shown in Figure 17. Damages near by the actuator (nearly in the plate centre) cannot be detected accurately. There are feature activations near by the edges and corners of the plate due to wave reflections, interferences and mode conversions (conversion of one Lamb wave mode into another mode). These artifacts disturb the damage prediction and localisation. Moreover, in this work a specimen structure consisting of only one composite

material is considered. Hybrid structures in terms of combined section regions of different materials, e.g., with intermediate stringers, will pose similar artifacts.

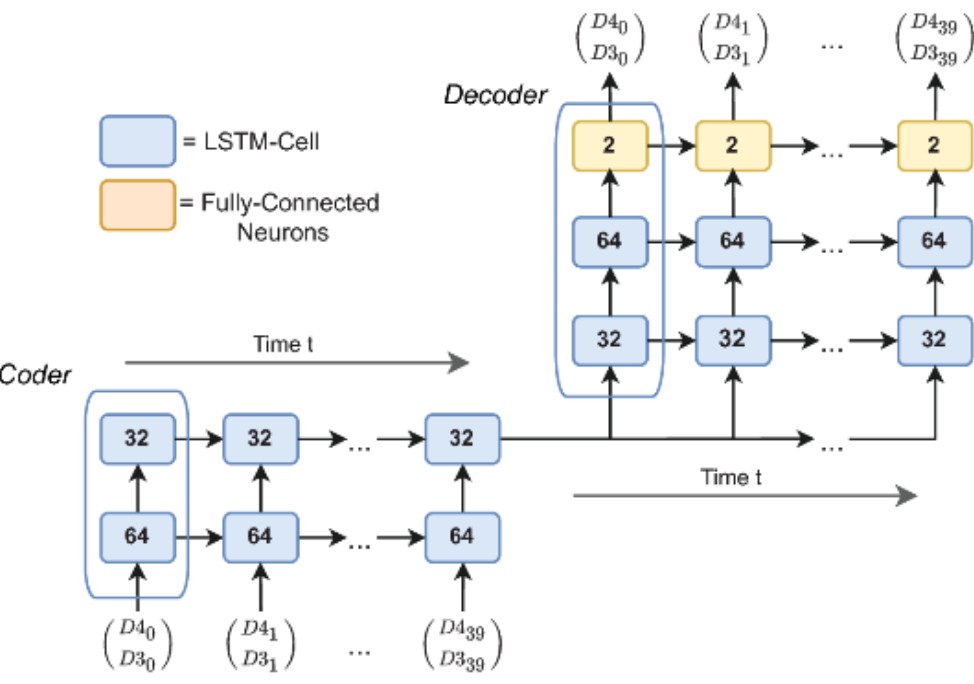

**Figure 15.** LSTM network architecture consisting of layers of LSTM cells: (Bottom) Encoder (TOP) Decoder; Time-unrolled and replicated presentation to illustrate the data flow on successive samples.

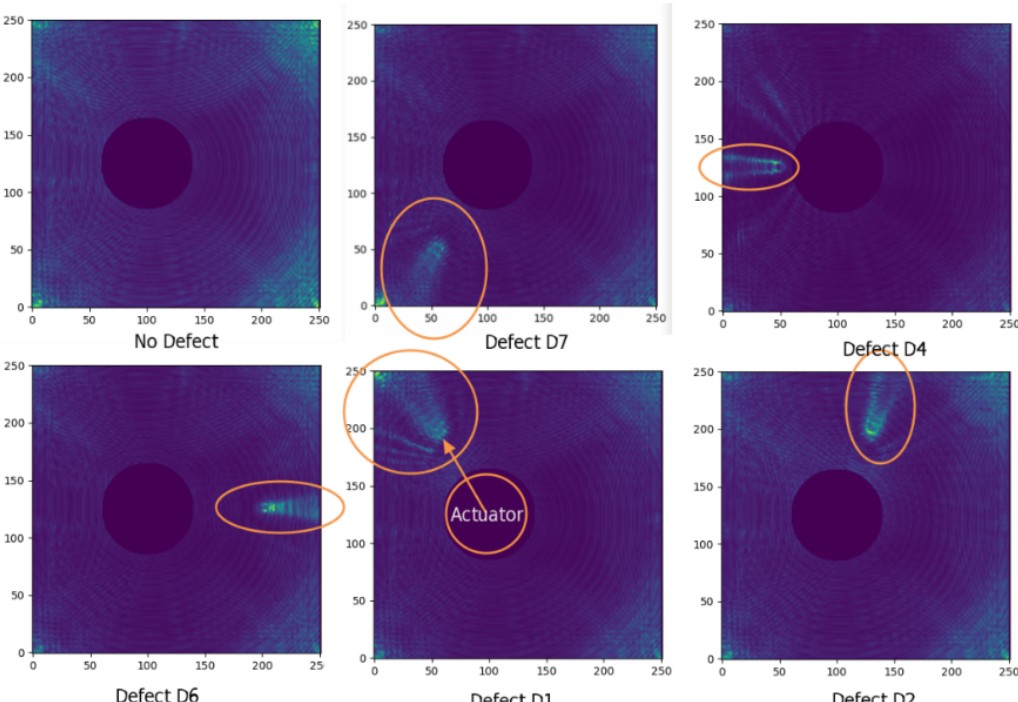

**Figure 16.** Typical examples of auto-encoder-based damage feature extraction. Shown are intensity images of the AE processed DWT features (i.e., each pixel of the image has a binary value and yellow colour indicates a detected anomaly); *x*- and *y*-axis in pixel coordinates, defect index is numerated from left to right and top to bottom (totally 9 defect positions).

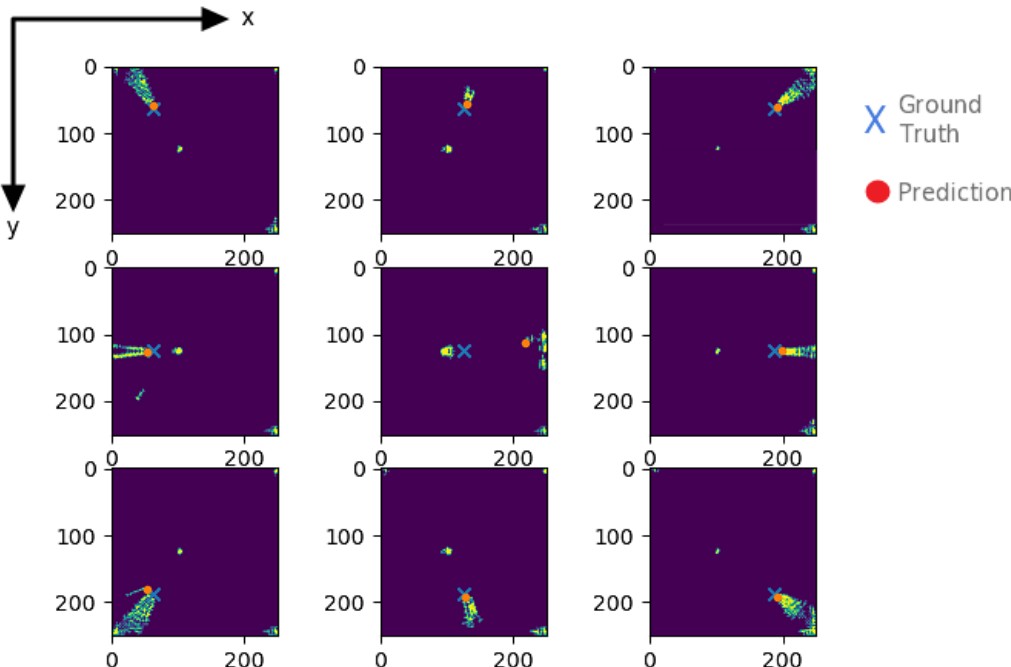

**Figure 17.** Spatial accuracy of the auto-encoder-based damage feature extraction with real damage positions markings (ground truth, blue cross) and the damage position approximated by DBSCAN (red circle); *x*- and *y*-axis in pixel coordinates.

*5.5. Results*

Results of the base-line approach using the auto-encoder output and density clustering feature extraction are shown in Figure 18. The accuracy measures are derived from all three data sets by Monte Carlo simulation adding Gaussian noise to the originally measures data sets. Some samples show false-negative predictions (indicated by black bars in the plot, typically 10–20% of the samples of one experiment). The training was only performed with the sensor data from the defect-less experiment. In contrast to the first local multi-instance learning, the global single-instance learning shows some false-negative predictions, i.e., no defect (position) was detected in case of an existing defect, indicated by black bars. The maximal prediction error occurs by a defect placed in the centre of the plate near by the actuator. The mean position error averaged over all sets and neglecting the three high error cases (D125-250, D250.250, and D250-375) is about 18 mm. The high position errors of the three aforementioned cases are a result of a low-contrast feature marking with fuzzy boundaries of the point clouds and high noise areas at the edges and corners of the plate due to wave interaction artifacts).

The following two box plots in Figure 19 show the dependency of the position accuracy on sensor noise (additive Gaussian noise added to the raw sensor signal) obtained by MCS. In both plots, the data are divided according to the individual noise levels. The first of the accumulated box plots summarises all experiments on all data sets. It can be seen that the median of all noise levels is slightly above an error of 20 mm, whereby with an SNR of 0 dB (same noise as signal strength) slightly higher errors occur. In the second grouped box plot the data was divided by noise levels, but also by the data sets. It can be seen that the largest error occurs in the second data set with VST defect mounting due to a weaker damage-wave interaction. Both plots also show outliers that differ significantly from the other points.

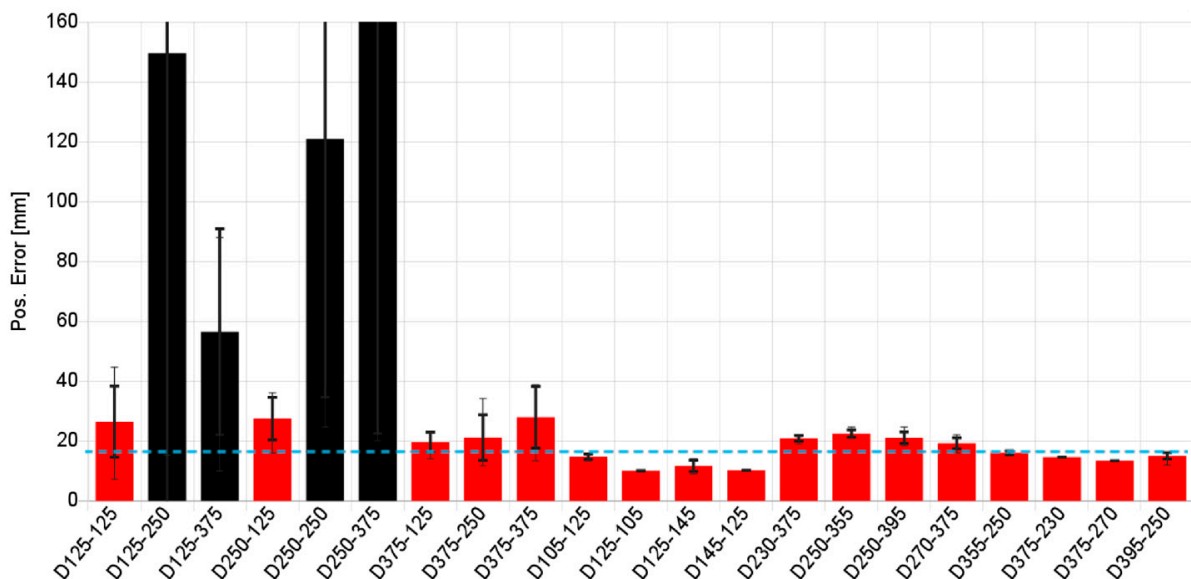

**Figure 18.** Accumulated prediction error statistics for all different defect positions and all data series (variance of 21 pseudo defect positions and two mounting technologies VST/MMA) [Label is defect position D⟨px⟩-⟨py⟩ in mm]. Black bars indicate false-negative prediction results for some of the data instances. The thick interval bars indicate err±σ, the thin lines the minimal and maximal error values.

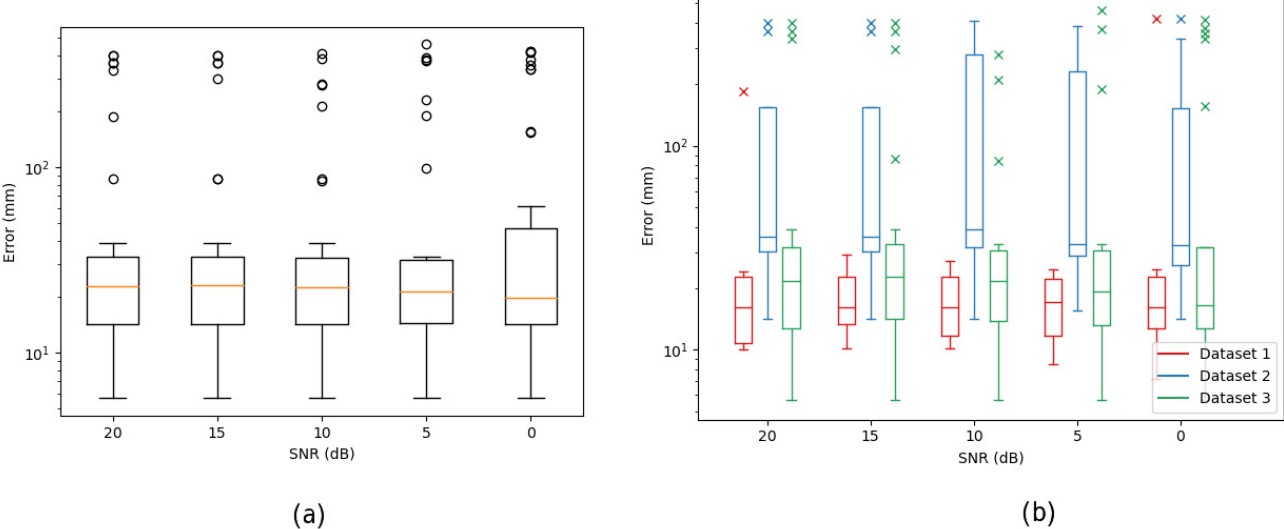

(a)                                                     (b)

**Figure 19.** Statistical analysis of the dependency of the position error on an artificial noise level added to the sensor signal. (**a**) Averaged over all data sets; (**b**) averaged over a specific data set. Single points are outliners.

By inserting noise, many previous unrecognised cases now result in quite large position errors, since noise is now recognised as a damage feature, which can be observed clearly with the second data set. With the other data sets, this is reflected in the outliers.

## 6. Comparison and Hybrid Architecture

### 6.1. Comparison of Both Methods

The damage diagnostics of both approaches presented in this work consists of the binary damage classification (i.e., there is a damage in a specimen or not) and the spatial damage localisation. The distributed multi-instance approach with global centre-of-mass fusion poses a high reliability with a true-positive and true-negative rate of 100%. The AE-based single-instance approach is affected by feature selection artifacts that result in

a false-positive rate of about 20% in some damage-cases (depending on the geometrical triangle sensor-actuator-damage with respect to the specimen boundaries). The averaged position accuracy is about 60 mm in the multi-instance and 20 mm in the single-instance approach, summarised in Table 2.

**Table 2.** Comparison of both damage diagnostics methodologies presented in this work. Annotations: 1. CPU Intel E5 2.9GHz 15MB L3, 2. RP3B, 1.2 GHz C4 ARM Cortex-A53 processor 512 KB L2, 3. GPU, 4. JSON data format, 5. nodejs 5.9 V8, 6. Binary data.

| Roperty | MTMP | STMP |
| :---: | :---: | :---: |
| Training Time | 1 min/node [1,5] | 1 h full image [3] |
| Inference Time | 10 ms/node [1,5], 70 ms/node [2,5] | 19 ms/pixel, 20 min/full image [3] |
| Model Size | 16 kB/node [4] | 272 kB [6]/full model |
| Damage Classification | 100% correct | 80% correct (20% false-neg.) |
| Damage Localisation | Avg. 60 mm | Avg. 20 mm |

The computational performance of both architectures is compared in Table 2 the computational performance is relevant for the deployment in distributed and embedded sensor networks as well as for real-time capability. The real-time capability is defined by the overall measuring time (e.g., 1 s) and the deadline for a result (that can be ranging from seconds to minutes).

The distributed multi-instance approach scales nearly linearly with the number of sensor nodes, and hence only one node is considered here. The single-instance approach was processed primarily on a GPU system. Even on an embedded computer such as the RaspBerry 3 with an ARM Cortex CPU the inference time is below 100 ms for one measurement and is suitable for real-time analysis. The multi-instance approach shows a comparable computational time on an embedded computer and by using a VM compared with the native code GPU-based algorithms per sensor. But the single-instance approach requires a higher sensor density for damage recognition and position estimation (at least $50 \times 50$ sensors).

Damage interaction with a MMA pseudo defect (at 80 kHz wave frequency) bases mainly on:

- Mode conversion $S_0$ to $A_0$ with minor amplitude
- Phase shift of $A_0$ mode with minor changes
- Amplitude reduction of $A_0$ mode (major change of first arrival)
- Scattering of $A_0$ mode
- Damage interaction with sealant pseudo defect at 80 kHz bases mainly on:
- Phase shift of $A_0$ mode
- Amplitude reduction of $A_0$ mode (major change of first arrival)
- Scattering of $A_0$ mode with minor amplitudes

Therefore, the ML algorithms detect mainly amplitude reduction (STMP) behind the defect, scattering (MTMP) and phase shift (STMP) of first arrival of the wave propagating from the actuator to the sensor. This is shown in Figure 20.

### 6.2. Hybrid Architecture

The unsupervised trained auto-encoder-based spatially generalised method poses on one hand a high accuracy, on the other hand an increased false-negative rates and in some damage cases a low accuracy. The supervised trained distributed multi-instance approach shows lower but reasonable accuracy with a zero false-negative and false-positive rate. Both methods can be fused to a hybrid architecture with improved performance:

- The MTMP instance approach is used for a first approximation of the damage location (region-of-interest marking, ROI) and a proper damage/no-damage classification

- The STMP instance approach uses the ROI and damage classification from the MTMP to discriminate inaccurate and wrong damage predictions (selective inference and feature extraction)

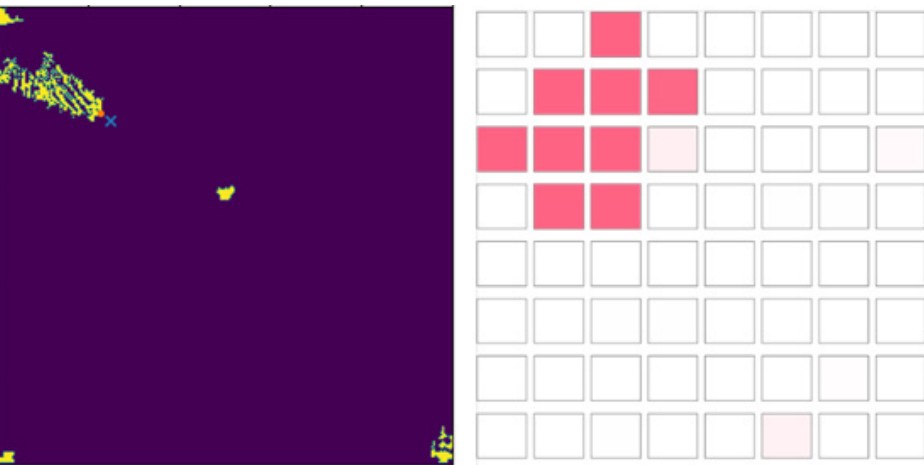

**Figure 20.** Comparison of output of the high-resolution STMP and low-resolution MTMP damage predictor functions (same experiment). STMP: single-instance learning of an auto-encoder with multi-instance prediction; MTMP: multi-instance learning with multi-instance prediction

## 7. Conclusions

Two different ML architectures were introduced that predict damages of a carbon fibre laminate plate with a high accuracy and reliability. Both approaches deliver a binary damage classification and an estimation of the damage location relative to the plate boundaries. The first is a low-, the second a high-resolution method with respect to sensor density and accuracy.

The first approach is a distributed multi-instance architecture with supervised training and suitable for the deployment in sensor networks. The sensor density is sparse (here $8 \times 8$ sensors). Each trained model instance is capable of predicting a damage in the neighbouring region around the sensor node. Global fusion finally approximates the spatial position of the damage achieving an average accuracy in the order of the sensor node distance (60 mm). The distributed approach showed 100% true-positive and 0% false-positive/negative damage classifications in all test data instances. The spatial graph of sensor, actuator, and damage and its position relative to the plate boundaries has an impact on the location prediction accuracy. The multi-instance models are bound to their spatial region where they are trained, thus they pose no spatial generalisation.

The second approach is a spatially generalised single-instance architecture with unsupervised training based on a base-line anomaly prediction using an auto-encoder. The single model instance can be replicated supporting multi-instance prediction. The sensor density is high (here $250 \times 250$ sensors). This approach showed an improved averaged accuracy in the order of tenth times of sensor distance (20 mm). This approach is not suitable for processing on embedded nodes of a sensor network due to high computational time and resource requirements (e.g., one GPU) and is considered as a laboratory diagnostics system and a reference analysis method. The main advantage of this approach is the unsupervised training method compared to the supervised first approach, avoiding labelling difficulties and a higher degree of generalisation (with respect to spatial, temporal, and environmental parameters).

Common to both architectures is a state-based recurrent ANN using a Long-short term memory cell processing feature transformed time-series data. Discrete wavelet decomposition is used as the primary feature transformation (the distributed multi-instance approach uses the first to fourth level, the auto-encoder approach uses the third and fourth decomposition level). The high-resolution approach delivered about 5% false-negative and

0% false-positive predictions. The false-negative rate can be dropped to zero by fusing and coupling both architectures. The binary damage classification is taken from the first system, the high-resolution position estimation by the second or by the first if the second system cannot find a damage.

There are still a lot of questions and evaluations to be done:

- Measurement and processing of more reference data with a broader range of different damage locations, mounting technologies, and environmental variations;
- Considering experiments with more than one damage (training and inference);
- Enhancing data augmentation beyond Monte Carlo simulation;
- Applying the methods to carbon fibre laminate plates with real impact damages;
- More rigorous investigation of the influence of sensor density, sensor failure, and sensor variations on prediction results;
- Implementing the distributed MTMP approach on a real sensor network with embedded low-resource computers.

**Author Contributions:** S.B. provided the multi-instance learning approach, the respective paper parts, and all analysis data; D.W. provided the single-instance learning approach, the respective paper parts, and all analysis data, and D.S. provided the measuring technique related parts and all experimental data. All authors have read and agreed to the published version of the manuscript.

**Funding:** The research was founded by the DFG (project number 418311604).

**Data Availability Statement:** Authors can confirm that all relevant data are included in the article.

**Conflicts of Interest:** The authors declare no conflict of interest.

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
