# Peer review of "Supervised Distributed Multi-Instance and Unsupervised Single-Instance Autoencoder Machine Learning for Damage Diagnostics with High-Dimensional Data—A Hybrid Approach and Comparison Study"

_computers, doi:10.3390/computers10030034_

Round 1
Reviewer 1 Report
The authors aim to predict (classify) and localize damaged regions of a carbon fiber laminate plate utilizing deep learning-based methods with "high accuracy and reliability". Authors are urged to address the following comments:
- “base-line” in the introduction should be “Base-line”; Section 2.1: “I.e” – a sentence cannot begin with “I.e.” like this; Section 7 – “capable to predict” – change to capable of predicting; “an” base-line – change to a base-line; “a sensor networks” – remove ‘a’; Section 2.3: “The test have to” – change to has to; Section 4.4: “Gloabal” to Global. Figure captions are off-center. Authors are urged to take note of such typographical, grammatical, and sentence construction errors, of which there are several in the manuscript, and correct them.
- Too many keywords have been presented – there should be no more than 5
- Section 2.1: Local Learning with Global Fusion – this section along with equation (3) does not make sense. Also, are d(t) I,j and di,j(t) same?
- Is the ensemble in equation 3 an intersection of individual learnt models? If so, how is that intersection performed?
- Compared to global learning, how are sensor interactions preserved in the local learning scheme?
- “analysis of 1. …..” - itemize this list sequentially on separate lines.
- Descriptive captions for all figures are necessary. For instance, what do the arrows in the sensor acquisition panel in Fig. 2 signify? Add descriptive captions for all figures.
- Fig. 6: How are relevant features containing signals segregated from other signals? The y-axis for Fig. 6(a) is not mentioned – is this sensor response? Also, figure caption labels are a mix of upper and lowercase characters.
- TP in Fig. 8 is not defined.
- Section 5.1: “a binary damage classificator” – explain what a “classificator” is; unless this is a typographical error. Fig. 14 shows a Fusion step and a threshold step – it is unclear what these steps accomplish and how they are beneficial for training the autoencoder.
- Section 5.2: How is 1:10 temporal down-sampling beneficial to the autoencoder training?
- initial “Fig. 17” on page 31 – this should be Fig. 16: There seem to be artifacts present in this figure – correct presence of such artifacts – in these sub-figures, there seem to be activations at the corners and edges – is there a specific reason behind this? – explain.
- “Fig. 17” on page 32: The legend – one of the legend keys – “Pradiktion” is in a language different from English
- Fig. 18: Explain high Pos. error for D250-250.
- Crucial discussions of the presented results and claims are missing from the manuscript. This should be done in a different section altogether.
- The accuracy and reliability angles of the proposed methods are not clearly discussed in the manuscript in its current form.
Author Response
Dear reviewer,
thank you for your numerous comments on our paper. The paper was extensively revised. A summary of changes is shown below.
## Changes:
1. Fig. 18 updated with statistically evaluated data from new experiments
2. Introduction was extended explaining AE methods
3. Numerous grammatical errors were fixed, language improved
## Reply to Review Report
The authors aim to predict (classify) and localize damaged regions of a carbon fiber laminate plate utilizing deep learning-based methods with "high accuracy and reliability". Authors are urged to address the following comments:
1. “base-line” in the introduction should be “Base-line”; Section 2.1: “I.e” – a sentence cannot begin with “I.e.” like this; Section 7 – “capable to predict” – change to capable of predicting; “an” base-line – change to a base-line; “a sensor networks” remove ‘a’; Section 2.3: “The test have to” – change to has to; Section 4.4: “Gloabal” to Global. Figure captions are off-center. Authors are urged to take note of such typographical, grammatical, and sentence construction errors, of which there are several in the manuscript, and correct them.
> Done
2. Too many keywords have been presented – there should be no more than 5
> The number of keywords was reduced to 5
3. Section 2.1: Local Learning with Global Fusion – this section along with equation (3) does not make sense. Also, are d(t) I,j and di,j(t) same?
> Paragraph, equation 3, and figure 1 revised with respect of notation; index confusion was fixed
4. Is the ensemble in equation 3 an intersection of individual learnt models? If so, how is that intersection performed?
> Yes, discussed in Sections 4 and 5 (spatial mass-of-center or density based fusion from binary classified local outputs)
5. Compared to global learning, how are sensor interactions preserved in the local learning scheme?
> There is no sensor interaction in terms of communication. But the distributed sensor signals are correlated by the wave propagation. A single sensor node processes only its local sensor and passes the pre-processed data to its local learning instance only predictiong the local state (e.g., a damage nearby).
6. “analysis of 1. …..” - itemize this list sequentially on separate lines.
> done
7. Descriptive captions for all figures are necessary. For instance, what do the arrows in the sensor acquisition panel in Fig. 2 signify? Add descriptive captions for all figures.
> All figure captions were extensively updated and extended with descriptive information
8. Fig. 6: How are relevant features containing signals segregated from other signals? The y-axis for Fig. 6(a) is not mentioned – is this sensor response? Also, figure caption labels are a mix of upper and lowercase characters.
> y-axis: amplitude of the US sensor signal; relevant features are selected by the DWT
9. TP in Fig. 8 is not defined.
> done
10. Section 5.1: “a binary damage classificator” – explain what a “classificator” is; unless this is a typographical error. Fig. 14 shows a Fusion step and a threshold step – it is unclear what these steps accomplish and how they are beneficial for training the autoencoder.
> Explanation added: "a binary damage classificator by applying a threshold function to the mean average error of the reconstructed signal and the originally measured signal"
11. Section 5.2: How is 1:10 temporal down-sampling beneficial to the autoencoder training?
12. initial “Fig. 17” on page 31 – this should be Fig. 16: There seem to be artifacts present in this figure – correct presence of such artifacts – in these sub-figures, there seem to be activations at the corners and edges – is there a specific reason behind this? – explain.
> fixed;
> There are feature activations near by the edges and corners of the plate due to wave reflections, interference, and mode conversion (with additional superpositon of different mode waves) at the plate-air boundary. These artifacts disturb the damage predicition and localisation. Moreover, in this work a specimen consisting of only one material is considered. Hybrid materials in terms of section regions of different materials will pose similar artifacts.
13. “Fig. 17” on page 32: The legend – one of the legend keys – “Pradiktion” is in a language different from English
> fixed
14. Fig. 18: Explain high Pos. error for D250-250.
> expl. added;
> The high position errors of the three aformentioned cases are a result of a low-contrast feature marking with fuzzy boundaries of the point clouds and high noise areas at the edges and corners of the plate 8due to wave interaction artifacts).
15. Crucial discussions of the presented results and claims are missing from the manuscript. This should be done in a different section altogether.
16. The accuracy and reliability angles of the proposed methods are not clearly discussed in the manuscript in its current form.
> Discussion was extended and both approache were compared and the quality discussed in Sec. "Comparison and Hybrid Architecture"
Reviewer 2 Report
The paper presents a very interesting work, in line with the needs of companies and researchers working with composite materials. The implementation of deep learning / machine techniques for fault detection or damage prediction are an asset for research improvements in the area. You are to be congratulated for presenting a challenging work where they can approach 2 techniques, supervised and unsupervised learning, each challenging in itself to implement and validate. The paper is well organized, written in a clear language and understood by the researchers working in the area.
At the end of paper says that were performed tests with raspberry pi. I understand that is a good platform for validation academic results, but not the best solution for industry.
Minor issues:
- Abstract - line 16: Where is written "spatially- and time-resolved" is "spatially".
- Subsection 2.4: The beginning of the section has no text content enough, perhaps you should just start in the next page.
- The same for section 3.
- The same on page 11 - "Removeable Pseudo Damages".
- Image quality must be improved (Fig.5, 17, 19).
- Figure 12 and 18: identify the x-axis.
- Figure 17 is repeated. On the second, write "prediction" in english.
- References needs to be formatted. Are written in different formats. Use the APA format.
Author Response
The paper presents a very interesting work, in line with the needs of companies and researchers working with composite materials. The implementation of deep learning / machine techniques for fault detection or damage prediction are an asset for research improvements in the area. You are to be congratulated for presenting a challenging work where they can approach 2 techniques, supervised and unsupervised learning, each challenging in itself to implement and validate. The paper is well organized, written in a clear language and understood by the researchers working in the area.
At the end of paper says that were performed tests with raspberry pi. I understand that is a good platform for validation academic results, but not the best solution for industry.
> The RP platform was only chosen as an example platform for the class of embedded computers and for some performance comparison. We will implement our algorithms on RP platform only for demonstration purposes.
Minor issues:
1. Abstract - line 16: Where is written "spatially- and time-resolved" is "spatially".
2. Subsection 2.4: The beginning of the section has no text content enough, perhaps you should just start in the next page.
3. The same for section 3.
4. The same on page 11 - "Removeable Pseudo Damages".
5. Image quality must be improved (Fig.5, 17, 19).
> Most figures were improved if possible
6. Figure 12 and 18: identify the x-axis.
> done/was added
7. Figure 17 is repeated. On the second, write "prediction" in english.
> fixed
8. References needs to be formatted. Are written in different formats. Use the APA format.
> All references were updated to the APA format.
Round 2
Reviewer 1 Report
The authors have addressed most of my comments satisfactorily. However, the manuscript still suffers from several English language and typographical errors - for instance, 'it's' being used in the wrong context - should be 'its' - for example, this can be seen in section 5.3. There are similar other mistakes as well.
Fig. 17 - legend cannot overlap figure - put legend sideways on a suitable place.
Fig. 18 - use a color other than black for the bars.
Author Response
All minor issues were fixed as suggested by the reviewer. Fig. 13 and 17 were updated and replaced. An additional Fig. (20) was finally added to show the differences of both propsed ML approaches more clearly and precisely.